# Induction of cross-neutralizing antibodies by a permuted hepatitis C virus glycoprotein nanoparticle vaccine candidate

Kwinten Sliepen [1,2] ✉, Laura Radić[1,2], Joan Capella-Pujol [1,2], Yasunori Watanabe[3], Ian Zon [1,2], Ana Chumbe [1,2], Wen-Hsin Lee [4], Marlon de Gast[1,2], Jelle Koopsen[1,2], Sylvie Koekkoek[1,2], Iván del Moral-Sánchez[1,2], Philip J. M. Brouwer[1,2], Rashmi Ravichandran[5,6], Gabriel Ozorowski [4], Neil P. King [5,6], Andrew B. Ward [4], Marit J. van Gils [1,2], Max Crispin [3], Janke Schinkel[1,2] & Rogier W. Sanders [1,2,7] ✉

Hepatitis C virus (HCV) infection affects approximately 58 million people and causes ~300,000 deaths yearly. The only target for HCV neutralizing antibodies is the highly sequence diverse E1E2 glycoprotein. Eliciting broadly neutralizing antibodies that recognize conserved cross-neutralizing epitopes is important for an effective HCV vaccine. However, most recombinant HCV glycoprotein vaccines, which usually include only E2, induce only weak neutralizing antibody responses. Here, we describe recombinant soluble E1E2 immunogens that were generated by permutation of the E1 and E2 subunits. We displayed the E2E1 immunogens on two-component nanoparticles and these nanoparticles induce significantly more potent neutralizing antibody responses than E2. Next, we generated mosaic nanoparticles co-displaying six different E2E1 immunogens. These mosaic E2E1 nanoparticles elicit significantly improved neutralization compared to monovalent E2E1 nanoparticles. These results provide a roadmap for the generation of an HCV vaccine that induces potent and broad neutralization.

The hepatitis C virus (HCV) causes a global epidemic with ~0.7% of the world population being infected. HCV is a blood-borne pathogen that can cause serious liver inflammation and it is responsible for around 300,000 deaths each year, rivaling the number of deaths caused by malaria[1]. The virus can persist for decades in an infected individual without causing symptoms, but the lingering infection can finally cause liver cirrhosis and/or liver cancer.

Direct-acting antiviral (DAA) therapy can effectively cure most HCV infections, but the large majority of infected individuals are unaware of their infection status or only receive treatment when too much liver damage has occurred. Moreover, therapy is costly and does not reach populations that are most at risk of acquiring HCV, such as injecting drug users and people living in low-income countries. As a consequence, the number of yearly new infections (1.4 million) exceeded the number of people receiving treatment (0.65 million) in 2020[2]. The development of an effective vaccine is critical for curtailing the virus. However, progress on the development of such a vaccine has been slow.

[1]Amsterdam UMC, location University of Amsterdam, Department of Medical Microbiology and Infection Prevention, Laboratory of Experimental Virology, Amsterdam, The Netherlands. [2]Amsterdam Institute for Infection and Immunity, Infectious Diseases, Amsterdam, The Netherlands. [3]School of Biological Sciences, University of Southampton, Southampton, UK. [4]Department of Structural Biology and Computational Biology, The Scripps Research Institute, La Jolla, CA, USA. [5]Department of Biochemistry, University of Washington, Seattle, USA. [6]Institute for Protein Design, University of Washington, Seattle, USA. [7]Department of Microbiology and Immunology, Weill Medical College of Cornell University, New York, USA. ✉e-mail: k.h.sliepen@amsterdamumc.nl; r.w.sanders@amsterdamumc.nl

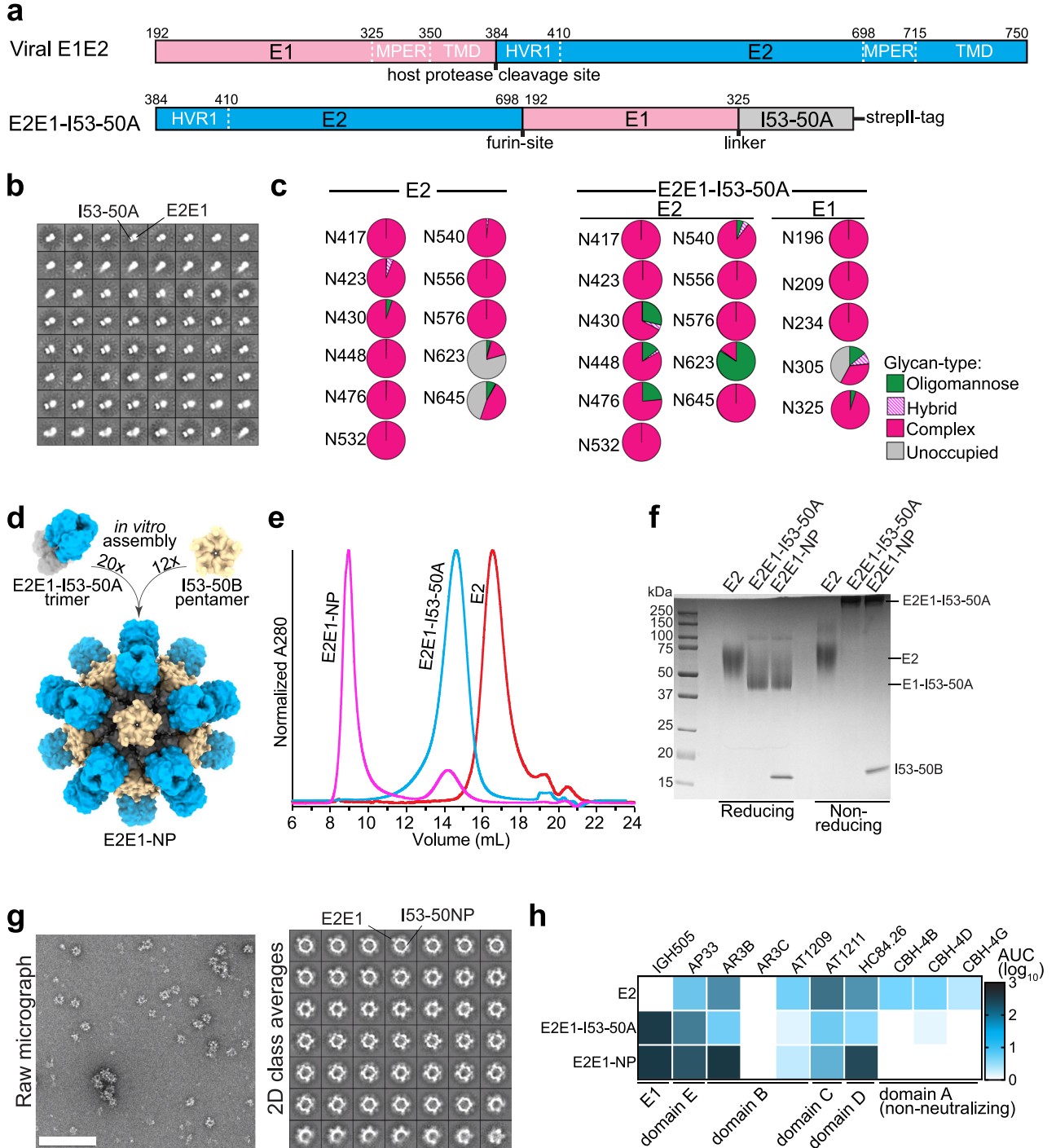

**Fig. 1 | Design and characterization of a recombinant HCV E2E1 nanoparticle immunogen. a** Top: linear representation of full-length E1E2 with the hypervariable region 1 (HVR1), transmembrane domains (TMD), and hydrophobic membrane-proximal external region (MPER) indicated. Bottom: the design of E2E1-I53-50A. The I53-50A.1NT1 (I53-50A) trimerization domain and E1 are separated by a Gly-Ser-rich linker (GSGGSGGSGGSGGS). The amino acid numbers are based on the standard H77 polyprotein numbering[104]. The amino acid sequences are shown in Fig. S2 and an overview of the other designs are depicted in Fig. S1a. **b** NS-EM 2D class averages of SEC-purified E2E1-I53-50A trimers. The E2E1 and I53-50A moieties are indicated. **c** Site-specific glycan analysis of E2 and E2E1-I53-50A produced in HEK293F cells and purified using Strep-Tactin followed by SEC. The relative occupancy and abundance of the different glycan species per potential *N*-glycosylation site (PNGS) are indicated in pie diagrams: non-occupied (gray), complex (magenta), hybrid (white/

magenta), and oligomannose glycans (green). **d** Schematic representation of the in vitro assembly of E2E1-I53-50 nanoparticles (E2E1-NP) by mixing E2E1-I53-50A trimers and the pentameric I53-50B subunits. **e** SEC profiles of E2E1-I53-50A trimers before assembly (blue) and post-assembly (magenta) in a Superose 6 column. E2 monomer is shown for comparison (red). **f** Reducing and non-reducing SDS gels of E2, E2E1-I53-50A, and E2E1-NP. Representative example of two experiments. **g** NS-EM images of E2E1-NPs with an example raw NS-EM image left and 2D class averages right. The white scale bar represents 200 nm. **h** Binding signal data derived from BLI (example curves depicted in Fig. S4a). Monoclonal antibodies (mAbs) were loaded on protein A sensors and binding signal was measured using an equimolar amount of E2 (100 nM). Depicted are the logarithms of the mean area under the curve (AUC) values from two independent experiments. Source data are provided as a Source data file.

A major roadblock for an HCV vaccine is the high sequence diversity of the virus, in particular in the envelope glycoproteins E1 and E2, which even exceeds the diversity of the human immunodeficiency virus 1 (HIV-1) envelope glycoprotein (Env)[3]. However, a subset of infected individuals develops broadly neutralizing antibodies (bNAbs) that neutralize a wide range of isolates and provide protection against HCV infection[4–6]. An efficient cellular immune response is likely required to clear an established infection[7,8], although some studies have suggested that bNAbs might also play a role in clearance[9–15]. Importantly, many of these HCV bNAbs require relatively little affinity maturation for their broad and potent neutralizing activity[9,12,16], suggesting that inducing such bNAbs by vaccination should be feasible.

All bNAbs target the E1E2 glycoprotein complex located on the outside of the virus. E1E2 is flexible and its intricate membrane-anchoring and the complex interplay between the E1 and E2 subunits have hampered vaccine development. Until recently[17], only high-resolution structures of the separate E1 and E2 subunits were available, but these did not provide the structural information to interpret the structure of E1E2[16,18–21]. As a consequence, most structure-based HCV vaccines efforts have been focused on the E2 alone, also because it is the main target of most known bNAbs[3,22–25]. Recently, we solved the structure of the native E1E2 heterodimer extracted from the cellular membrane[17]. Previous studies have suggested that E1E2 might also form trimers of heterodimers on the virion[26–28].

Many candidate HCV recombinant glycoprotein vaccines contain E2 only[24,29–33]. Although some vaccination experiments with recombinant E2 yielded NAbs that neutralized the autologous virus and some heterologous strains[24,32], other E2 immunogens did not even elicit autologous neutralization[29,31]. This inconsistency is probably caused by differences in neutralization assays, animal model used, and the nature of the vaccine[34]. Nanoparticle display, and stabilizing mutations can moderately improve E2 immunogenicity[22,23,25,35], but recombinant E1E2 has not shown considerable improvement over E2 so far[36–38]. Other vaccine strategies, such as inactivated HCV or cocktails of virus-like particles have shown some promise but are difficult to produce and/or induce only weak NAb responses[39,40].

Here, we present the generation of soluble E2E1 trimers for HCV vaccine design by permutation of the genes for the E1 and E2 subunit. These resulting E2E1 trimers could be displayed on two-component nanoparticles. We exploited this immunogen design for generating nanoparticle cocktails and mosaic nanoparticles that co-display several E2E1 trimers. These mosaic nanoparticles induced significant neutralizing breadth after only two immunizations.

## Results

### Permutation of E1E2 and fusion to I53-50A generate E2E1 trimers

Several studies have suggested that E1E2 on virions consists of trimers of E1E2 dimers[26,27]. We set out to design a recombinant form of E1E2 that mimics this trimeric configuration. Our initial designs were based on HCV isolate AMS0232, which is a relatively neutralization-resistant primary genotype 1a strain isolated in the Netherlands from a MOSAIC cohort participant[17,41]. Several AR3-targeting bNAbs, such as AR3C and AR3B, weakly neutralize AMS0232 probably because this virus contains relatively rare mutations in antigenic region 3 (AR3): T431 (~65% D; ~1% T in natural HCV isolates), E434 (~44% N; ~5% E) and I442 (~78% F; ~6% I)[17]. Therefore, we introduced an I442F change in our recombinant AMS0232 glycoprotein designs. We truncated E1 at position 325 and E2 at position 698 to remove the TMD and the predicted membrane-proximal external regions (MPER), which could cause aggregation due to their hydrophobicity (Fig. S1a)[42]. E1 and E2 were placed in tandem (E1E2) and the subunits were separated by an optimized furin cleavage site consisting of six arginines (R6) (E1E2 in Fig. S1b)[43]. At the start of this project, alanine-scanning studies of bNAb epitopes suggested that the N-terminus of E1 is adjacent to the C-terminus of E2[4,44]. Therefore, we aimed to bring these regions in closer proximity by permutation,

sometimes referred to as circular permutation[45], of E1 and E2 (E2E1) (Fig. S1b). More recently we succeeded in solving the structure of membrane-bound E1E2[17]. This structure reveals that several amino acids in the E2 C-terminus (the region from R659 to I690) and E1 N-terminus (Y192-N205) establish crucial interactions in the E1/E2 interface. Furthermore, D698 and N325 that represent the C- and N-termini of E2 and E1 in the E2E1 design are in close proximity according to the E1E2 structure (~18 Å) (Fig. S1c). The structure thus provided further post hoc justification for our rationale to permute E1 and E2 in our recombinant HCV glycoprotein designs. The TMD of E1 trimerizes E1E2 on the virion[26]. The permuted design allowed us to mimic the trimerization functionality of the E1 TMD by fusing the I53-50A trimerization domain to the C-terminus of E1 (E2E1-I53-50A)[46]. This also increases antigen valency which could enhance B cell receptor activation, while the trimeric conformation might also shield non-neutralizing epitopes, such as the membrane-facing side of E1E2 (Fig. 1a and Fig. S1b)[17]. I53-50A also allows assembly of two-component icosahedral nanoparticles for multivalent display of the antigen[46–50]. Recombinant E2 was also truncated at position 698. The amino acid sequences of these constructs are listed in Fig. S2.

The different HCV constructs were transfected in HEK293F cells and the proteins purified using Strep-Tactin-based purification followed by size-exclusion chromatography (SEC). E1E2 and E2E1 produced oligomers and smaller species, presumably heterodimers ("peak 1" and "peak 2" in Fig. S1d, respectively). In contrast, E2E1-I53-50A eluted within a narrow peak that corresponds to trimers (Fig. S1d). Reducing SDS-PAGE gels revealed efficient cleavage of the junction between E1 and E2 in the E1E2, E2E1, and E2E1-I53-50A proteins (Fig. S1e, left). Non-reducing SDS-PAGE showed long smears for the oligomeric species of E1E2 and E2E1, indicating the presence of heterogeneous disulfide formation[31], while E2E1-I53-50A showed one distinct band larger than 250 kDa, suggesting that E2E1-I53-50A forms a disulfide-bonded trimer (Fig. S1e, right).

E2E1-I53-50A engaged several HCV bNAbs targeting different epitopes, including AP33 (antigenic site 412 (AS412)/domain E), AR3B (antigenic region 3 (AR3)/domain B), AT1211 (domain C), HC84.26 (domain D), all located on E2, as well as IGH505 which targets an epitope on E1, but not AR4A and AT1618, which target conformational epitopes that require natively folded E1E2 (Fig. S1f)[4–6,13,17,51–53]. Negative-stain electron microscopy (NS-EM) of the E2E1-I53-50A trimer revealed that E2E1 and I53-50A are visible as two separate moieties of which E2E1 was visible as a more diffuse density, which is explained by the relatively long linker separating E2E1 from I53-50A and the inherent flexibility of the HCV glycoprotein[54,55]. Native membrane-bound E1E2 is also heterogeneous and it required the addition of three bNAbs, including AR4A, to solve its structure[17], which further illustrates that flexibility and heterogeneity are intrinsic features of the HCV glycoprotein. Because of this flexibility we were not able to reconstruct a higher-resolution model of E2E1.

### The glycan shield of E2E1-I53-50A is more complete than that of E2

Glycans play an important role in the folding and immunogenicity of HCV glycoproteins[56]. We used hydrophilic interaction liquid chromatography-ultra performance liquid chromatography (HILIC-UPLC) to determine the overall glycan composition of recombinant E2 and E2E1-I53-50A[57]. First, glycans were released using PNGase F, fluorescently labeled, and measured by HILIC-UPLC. To determine the abundance of oligomannose glycans, the labeled glycans were sequentially digested by Endo H and then measured by HILIC-UPLC. E2 and E2E1-I53-50A contained mostly highly processed complex glycans with a slight increase in oligomannose content for E2E1-I53-50A compared to E2 (Fig. S3a). To compare the glycan types and occupancy of the individual potential N-glycosylation sites (PNGS) of E2 monomers and E2E1-I53-50A, we used site-specific glycosylation analysis[17,58,59].

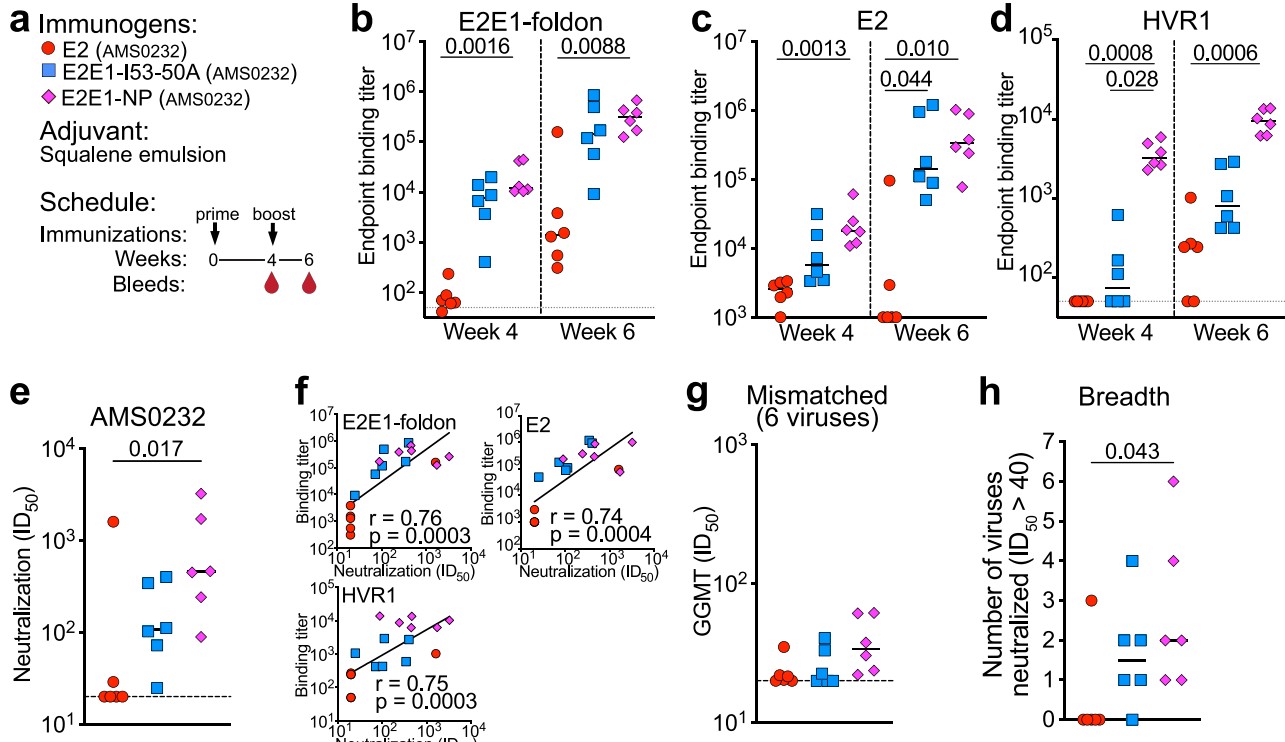

**Fig. 2 | Immunogenicity of E2E1-NPs in rabbits. a** Rabbit immunization schedule. Six rabbits were immunized with E2 monomers, E2E1-I53-50A trimers, or E2E1-NP (10 μg E2, or equimolar amount of E2) at weeks 0 and 4. Bleeds were taken at weeks 4 and 6. All immunogens are based on the genotype 1a AMS0232 strain. **b**–**d** Rabbit serum binding titers to E2E1-foldon trimers (**b**), E2 monomers (**c**), and HVR1 (first 27 amino acids of AMS0232 E2) (**d**). **e** Neutralization titers measured against the sequence-matched AMS0232 HCVpp. **f** Correlations between autologous neutralization (**e**) and binding titers (**b**–**d**). Spearman $r$ and $p$-values (two-tailed) are indicated. **g** Global geometric mean titers (GGMT) were calculated based on the neutralization $ID_{50}$ titers against six HCVpp strains. Each dot represents a single rabbit serum. Neutralization $ID_{50}$ titers of the individual viruses are depicted in Table S1. **h** Breadth of the neutralizing response defined as the number of HCVpp strains neutralized with an $ID_{50}$ titer above 40. Horizontal lines in (**b**)–(**e**), (**g**), (**h**) indicate the median values. Significant differences between groups in (**b**)–(**e**), (**g**), (**h**) were determined using a Kruskal–Wallis test followed by Dunn's post-test (**b**–**e**, **g**, **h**); $n = 6$ rabbit sera per group. $P$-values for significant differences are indicated on top of the graphs. Source data are provided as a Source data file.

Most PNGS on the E2 monomer were fully occupied by highly processed complex glycans (Fig. 1c and Fig. S3b), except PNGS at 623 and 645, which were only partially occupied (~20% and ~55%, respectively) (Fig. 1c). In contrast, on E2E1-I53-50A all PNGS in E2 were fully occupied, mostly by complex glycan species. E2E1-I53-50A contained elevated levels of oligomannose at positions 430, 448, 476, and 623 possibly reflecting steric occlusion of these sites to mannosidase because of the trimeric organization of E2E1. Four out of the five glycans on E1 were fully occupied with predominantly complex-type glycans, while the 305 PNGS was frequently unoccupied (~40%), and when occupied contained a mix of oligomannose (~12%), hybrid (5%), or complex glycans (~30%). N325 is usually unoccupied in native E1E2, but is occupied in E2E1-I53-50A[17,60]. Overall, E2E1-I53-50A is almost fully glycosylated and contains a glycan profile that resembles that of E1E2 on HCV pseudoviruses[61].

**E2E1-I53-50A trimers assemble into nanoparticles efficiently**

To assemble E2E1-I53-50A into nanoparticles (E2E1-NP), we mixed the purified E2E1-I53-50A trimers with the pentameric I53-50B.4PT1 and we used SEC to separate the expected higher molecular weight nanoparticles from the E2E1-I53-50A trimers[46] (Fig. 1d). Indeed, the mixture of E2E1-I53-50A with I53-50B.4PT1 eluted substantially earlier from the Superose 6 column than E2E1-I53-50A trimer alone indicating successful nanoparticle formation (~9.0 mL versus ~14.5 mL for E2E1-I53-50A) (Fig. 1e). Reducing SDS-PAGE confirmed that E2E1-I53-50A and I53-50B.4PT1 were present in the higher molecular weight peak that represents the nanoparticle fraction (Fig. 1f). NS-EM images of the SEC-purified complexes showed that the nanoparticles are ~35 nm in

diameter and in the 2D class averages the I53-50 nanoparticle core and protruding E2E1 densities were clearly visible (Fig. 1g). Next, we used bio-layer interferometry (BLI) to compare the antigenicity of E2E1-NPs, E2E1-I53-50A trimers and E2 monomers (Fig. 1h and Supplementary Fig. 4a). The nanoparticles displayed enhanced binding signals for bNAbs AR3B, HC84.26 and AP33 compared to E2E1-I53-50A trimers, possibly because of the increased avidity of these bNAb epitopes. In contrast, narrowly neutralizing AT1211, which target domain C, and non-neutralizing antibodies CBH-4B, CBH-4D, and CBH-4G engaged E2 monomers more efficiently. As expected from the poor neutralizing activity of AR3C against AMS0232 HCVpp[17] and the mutations present in the AR3C binding site outlined above, none of the recombinant glycoproteins bound to AR3C. Together, these results demonstrate that several bNAb epitopes are accessible on self-assembling two-component E2E1-NPs.

**E2E1-NPs induce potent neutralizing antibody responses in rabbits**

To assess the immunogenicity of E2E1-NPs, we immunized rabbits twice, at weeks 0 and week 4, with 10 μg E2 monomers, 22 μg E2E1-I53-50A trimers or 27 μg E2E1-NPs, resulting in equimolar amounts of E2, adjuvanted in squalene emulsion (Fig. 2a). Antibody responses were measured at weeks 4 and 6. To assess anti-E2E1 trimer responses without measuring Ab responses against the I53-50A domain, we generated an E2E1 trimer containing a foldon trimerization domain with a His-tag (E2E1-foldon) (Supplemental Fig. 4b). After the prime, binding titers to E2E1-foldon were ~180-fold higher for the E2E1-NP immunized rabbits compared to rabbits that received E2 (median titer

of 12,190 for E2E1-NP versus 67 for E2, $p = 0.0016$; Fig. 2b), and only slightly higher compared to the titers elicited by the E2E1-I53-50A trimers (median titer 7586; $p = 0.84$). After the boost, binding titers mounted by E2E1-NP were still more than 200-fold higher than those induced by E2 monomers (319,153 versus 1419, $p = 0.0088$; Fig. 2b). Similar results were obtained when we measured binding titers to E2 monomers (Fig. 2c) with higher anti-E2 binding titers measured at week 6 for E2E1-I53-50A trimer ($p = 0.044$) and E2E1-NP ($p = 0.01$) immunized rabbits compared to E2 immunized rabbits.

The hypervariable region 1 (HVR1) is a highly sequence-variable immunodominant epitope located at the N-terminus of E2. The HVR1 drives early strain-specific Ab responses that neutralize the autologous virus, but usually do not display breadth[62]. The HVR1 sequence differs greatly between strains and its sequence evolves during HCV infection. It has been hypothesized that the HVR1 distracts from inducing responses against conserved cross-reactive epitopes[62,63]. We used a peptide based on the HVR1 of AMS0232 to measure HVR1 responses in the sera of these rabbits. E2E1-NP mounted strong binding responses against the HVR1 after the prime, while none of the rabbits immunized with E2 and only three out of six rabbits receiving E2E1-I53-50A raised measurable HVR1 binding titers ($p = 0.0008$ and $p = 0.028$, respectively, for the comparison) (Fig. 2d). After the boost, four out of six rabbits immunized with E2 displayed HVR1 binding titers, but the difference with E2E1-NP immunized rabbits was still large (38-fold) and significant ($p = 0.0006$), while HVR1 binding was now detected in all E2E1-I53-50A trimer immunized rabbits (Fig. 2d).

All week 6 sera from rabbits immunized with E2E1-NP and only one serum from E2-immunized rabbit neutralized the sequence-matched AMS0232 strain (median $ID_{50}$ of 457 for E2E1-NP versus median $ID_{50}$ of 20 for E2, $p = 0.0135$; Fig. 2e and Fig. S5). Five out of six rabbits receiving E2E1-I53-50A trimers neutralized AMS0232 HCVpp (median $ID_{50}$ of 107, not significantly different from the E2 or E2E1-NP groups). The pre-immunization sera did not neutralize AMS0232 and the week 6 sera did not neutralize the VSV-G pseudovirus demonstrating that the measured titers are not due to aspecific neutralizing activity in the sera (Fig. S5). Neutralization titers correlated strongly with binding titers to E2E1-foldon, E2, and HVR1 ($r = 0.75, 0.74$, and $0.76$, respectively; $p = 0.0003, 0.0004, 0.0003$, respectively; Fig. 2f). We next measured neutralization against six mismatched viruses (Table S1). Overall, heterologous neutralization was detectable albeit at low level. The global geomean titer (GGMT) against the mismatched viruses was higher for the E2E1-NP immunized rabbits, but differences were not statistically significant (median GGMT of 34 for E2E1-NP versus 21 and 21 for E2 and E2E1-I53-50A, $p = 0.08$ and $p = 0.18$, respectively). However, the number of viruses neutralized ($ID_{50} > 40$) by animals that received E2E1-NP was significantly higher than animals that received E2 monomers (Fig. 2h) ($p = 0.04$ for the comparison). In fact, only one E2 recipient neutralized more than one virus, while four out of six animals from the E2E1-NP groups neutralized at least two viruses. We also observed a consistent but non-significant trend indicating that E2E1-NP elicits higher binding and neutralizing Abs compared to E2E1-I53-50A trimers (Fig. 2b–h). In summary, AMS0232-based E2E1-NPs elicited more potent binding and neutralizing antibodies than E2 and these responses target a broader range of viruses.

## The E2E1-NP design is generalizable across HCV genotypes
Next, we wanted to explore the applicability of the E2E1-I53-50A design for producing nanoparticle immunogens based on a different genotype. A recent study suggested that HCV strains from genotype 3 mount more favorable Ab responses than strains from other genotypes[64]. Therefore, we next generated E2, E2E1-I53-50A and E2E1-NP based on the genotype 3 AMS3a strain isolated from a chronically infected individual in the Amsterdam Cohort Studies[13]. The AMS3a E2E1-I53-50A protein was purified via Strep-Tactin affinity purification and the trimers eluted at the same column volume as AMS0232 E2E1-

I53-50A (Fig. 3a). The AMS3a E2E1-I53-50A assembled efficiently as nanoparticles after mixing with I53-50B.4PT1 as assessed by SEC and NS-EM (Fig. 3a, b).

We then compared the immunogenicity of AMS3a-based E2, E2E1-I53-50A, and E2E1-NP immunogens. As in the previous experiment, rabbits were immunized twice with equimolar amounts of E2 (Fig. 3c) and the responses were measured at week 6, i.e., two weeks after the boost. Rabbits immunized with E2E1-NP elicited 18-fold higher binding titers against E2E1-foldon than E2 ($p = 0.0029$), but were not higher compared to E2E1-I53-50A immunized rabbits (Fig. 3d). The binding titers against AMS3a HVR1 seemed higher for the E2E1-NP immunized rabbits, but this difference was not significant ($p = 0.105$) (Fig. 3e). We used competition ELISA to determine if sera of the rabbits targeted known bNAb epitopes (Fig. 3f). The sera from all three groups competed most efficiently with AP33, targeting domain E, which is located at the C-terminus of the HVR1. Interestingly, sera from rabbits immunized with E2E1-NP competed more efficiently with AT1209 and AR3C than sera from E2-immunized rabbits ($p = 0.007$ and $p = 0.039$, respectively). These bNAbs engage domain B, which overlaps with the CD81 binding site and is an important target site for vaccine design, because it is a conserved neutralizing epitope and its protein surface is relatively exposed[17].

Five out of the six rabbits immunized with E2E1-NP mounted neutralizing antibodies against the sequence-matched AMS3a virus. The median neutralization titer of AMS3a E2E1-NPs immunized rabbits were higher than from animals receiving E2 or E2E1-I53-50A trimer but the differences did not reach statistical significance (Fig. 3g). Aggregating the immunogenicity results of this AMS3a-based study (Fig. 3) and the AMS0232-based study (Fig. 2) shows that E2E1-NPs elicit ~10-fold higher NAb titers ($p = 0.0012$), ~60-fold higher E2E1-foldon binding titers ($p < 0.0001$) and ~25-fold higher HVR1 binding titers ($p < 0.0001$) compared to E2 (Fig. S6). In the same comparison, we observed a trend that suggested E2E1-I53-50A trimers elicit lower binding and neutralizing titers compared to E2E1-NP (non-significant), but higher titers than the E2 groups (non-significant, except for E2E1-foldon binding ($p = 0.046$)) (Fig. S6). We also tested the breadth of the sera against an extended panel of 15 additional viruses (Table S1). Overall, NAb breadth was limited, but again we observed that E2E1-NPs elicited a broader and more potent NAb response than E2 when comparing the GGMT of the 16 viruses (median GGMT of 37 for E2E1-NPs versus 27 for E2, $p = 0.048$). In general, rabbits receiving E2E1-NP neutralized more viruses (three to nine strains) than E2 recipients (one to five strains) (Fig. 3i). Interestingly, increased neutralization breadth correlated with stronger competition of AP33 ($p = 0.0093$; $r = -0.59$) and AR3C ($p = 0.04$; $r = -0.48$), suggesting that serum Abs targeting these epitopes contribute to neutralization breadth (Fig. 3j). Together, these immunogenicity results confirmed that E2E1-NPs are superior to soluble E2 for eliciting NAbs.

## Multiple E2E1-I53-50A trimers can be assembled into mosaic nanoparticles
The two E2E1-NPs tested above, i.e., those based on the genotype 1a strain AMS0232 and the genotype 3 strain AMS3a, induce autologous neutralization, but the NAb breadth of the sera was limited. One complicating factor might be the immunodominant HVR1, which probably functions as an immunological decoy, and induces strong strain-specific NAbs but prevents the induction of bNAbs[63]. We hypothesized that co-display of multiple E2E1 trimers on the same nanoparticle might help to elicit broader responses. Co-display can efficiently focus responses towards cross-neutralizing epitopes that are conserved among the displayed antigens, because the cross-reactive epitopes benefit most from the multivalent display[65,66]. Second, co-display in effect decreases the avidity of strain-specific epitopes, such as the HVR1, that are then presented on only a few of the displayed antigens.

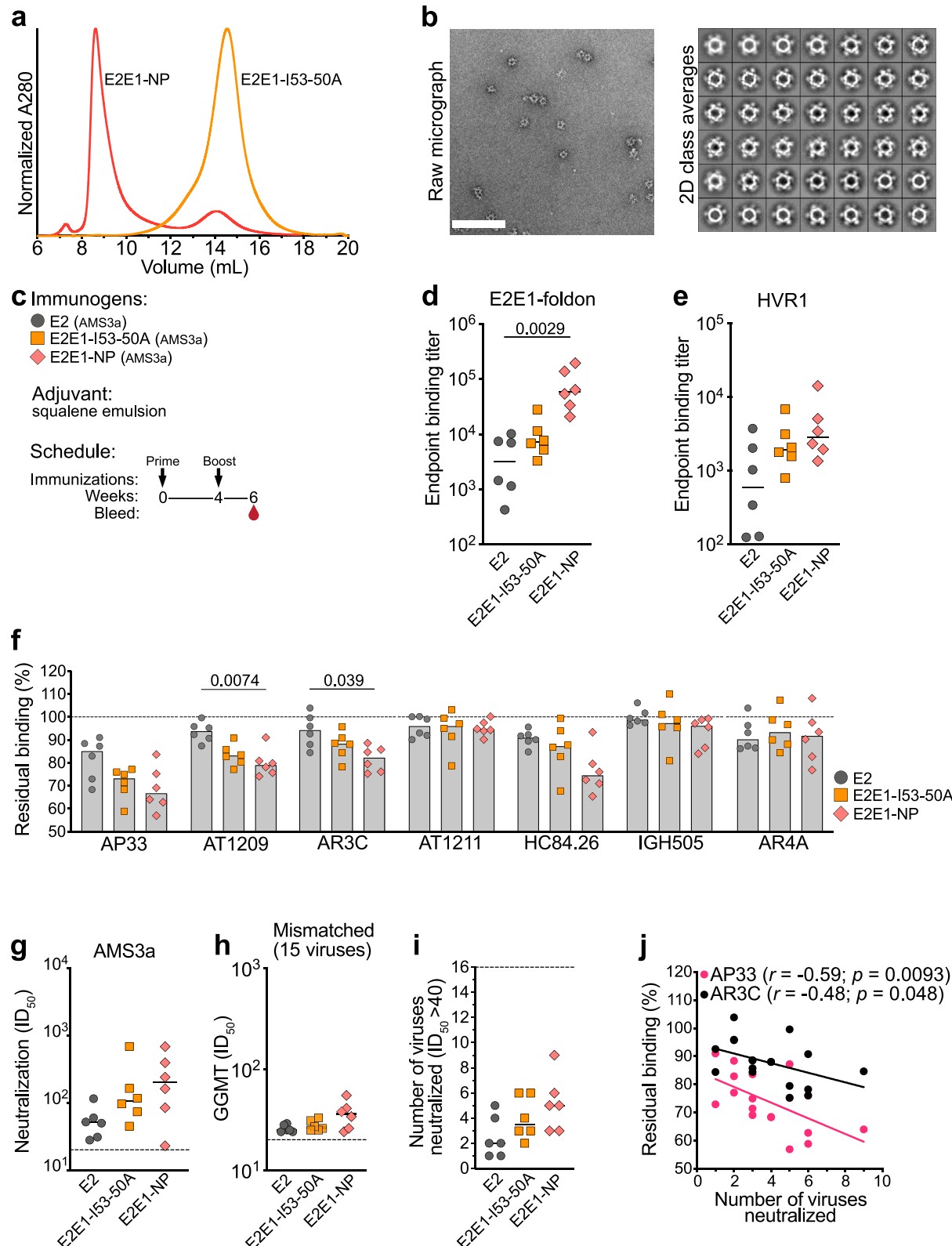

We set out to generate mosaic nanoparticles that co-display different E2E1-I53-50A trimers, in particular ones from different genotypes and with divergent HVR1 domains. We therefore applied the following criteria for the sequence selection. First, we picked sequences from each of the six major genotypes (Fig. S7a). Second, we selected sequences that displayed prominent divergence in their HVR1 sequence (Fig. S7b). Considering these points, we chose

AMS3a and six other E1E2 sequences, representing at least one sequence from each of the six major genotypes (Fig. S7a, b) and generated E2E1-I53-50A constructs from them as described above. Following purification, the six new proteins behaved similarly in SEC and Blue Native PAGE analyses as the AMS0232 and AMS3a E2E1-I53-50A proteins (Fig. S8a, b, respectively). All proteins were cleaved efficiently by furin, except for UKNP2.2.1 E2E1-I53-50A

**Fig. 3 | Characterization and immunogenicity of an E2E1-NP based on a genotype 3 strain. a** SEC profiles from a Superose 6 column of AMS3a E2E1-I53-50A and AMS3a E2E1-NPs. **b** NS-EM images and 2D class averages of AMS3a E2E1-NPs. The white scale bar represents 200 nm. **c** Rabbits were immunized at weeks 0 and 4 with E2, E2E1-I53-50A, and E2E1-NP, all based on the AMS3a strain. Antibody responses were measured at week 6. **d** Binding titers to E2E1-foldon (AMS3a) and **e** HVR1 peptide (AMS3a). **f** Competition ELISA of immunized rabbit sera. Residual binding of mAbs was measured after incubating rabbit sera with AMS3a-derived E1E2 from cell lysates. Each dot represents the value of three independent experiments measured in duplo or triplo. Bars indicate the median values. **g** Neutralization titers against the sequence-matched AMS3a HCVpp. **h** Global geometric mean titers

(GGMT) were calculated based on the neutralization $ID_{50}$ titers against fifteen non-matched HCVpp strains. Each dot represents a single rabbit serum. Neutralization $ID_{50}$ titers of the individual viruses are depicted in Table S1. **i** Breadth of the neutralizing response defined as the number of HCVpp strains neutralized with an $ID_{50}$ titer above 40. **j** Correlations between breadth of the serum response (**i**) and residual binding measured in competition ELISA (**f**). Spearman $r$ and $p$-value (two-tailed) are indicated. Horizontal lines in (**d**), (**e**), (**g**)–(**i**) indicate the medians values. Significant differences between groups in (**d**)–(**i**) were determined using Kruskal–Wallis test followed by Dunn's post-test, $n = 6$ rabbit sera per group. $P$-values for significant differences are indicated on top of the graphs. Source data are provided as a Source data file.

(Fig. S8c), and all constructs formed disulfide-linked trimers (Fig. S8d).

We mixed equimolar amounts of the six well-cleaved E2E1-I53-50A trimers (H77, AMS2b, AMS3a, UKNP4.1.1, UKNP5.2.1, UKNP6.1.2) with the appropriate amount of the pentameric B component to generate a hexavalent mosaic E2E1-NP (Fig. 4a, b). We used the same E2E1-I53-50A trimers to assemble the monovalent nanoparticles (Fig. 4b). The mosaic and monovalent E2E1-NPs displayed similar morphology as the AMS0232 and AMS3a E2E1-NPs when assessed by NS-EM (Fig. 4c; Fig. S8e). The E2E1-NPs displayed favorable antigenicity with overall decent binding to bNAbs and virtually no binding to the CBH-4B non-NAb (Fig. 4d and Fig. S9).

To determine whether mosaic nanoparticles are capable of activating bNAb-carrying B cells, we generated a B cell line carrying the AR3C bNAb on its cell surface to assess B cell activation by measuring calcium flux. We compared AR3C B cell stimulation by mosaic E2E1-NPs with UKNP4.1.1 E2E1-NPs, because these nanoparticles showed similar binding to AR3C (Fig. 4d and Fig. S9). Adding UKNP4.1.1 E2E1-NPs or the mosaic E2E1-NPs resulted in a calcium influx indicative of B cell activation, while equimolar UKNP4.1.1 E2E1-I53-50A trimers activated the B cells less efficiently (Fig. 4e). These results show that E2E1-NPs activate B cells more efficiently than single trimers, and that the mosaic particles efficiently activated B cells that carry a broadly reactive B cell receptor.

## Mosaic E2E1-NPs induce cross-neutralizing antibody responses

We compared the immunogenicity of mosaic E2E1-NPs to a vaccine consisting of a cocktail of E2E1-NPs carrying the same E2E1-I53-50A trimers, as well as to the AMS3a E2E1-NPs described above (Fig. 5a). The rabbits were immunized twice as per the scheme above and received a total of 27 µg E2E1-NPs. Thus, the cocktail-immunized rabbits received a mix containing $6 \times 4.5$ µg E2E1-NPs. First, we tested serum binding to AMS3a E2E1-foldon and the AMS3a HVR1 and observed that the monovalent AMS3a E2E1-NPs elicited slightly higher AMS3a-specific binding titers than the cocktail and mosaic nanoparticle groups, but these differences were not significant (Fig. S10). Next, we used a hexavalent mix of sequence-matched E2E1-foldon trimers and a hexavalent mix of HVR1 peptides to measure the binding antibody responses after two immunizations. Binding titers against E2E1-foldon binding titers did not differ between the three groups (Fig. 5b). The cocktail E2E1-NP vaccine induced ~4.5-fold higher anti-HVR1 binding titers than the other two vaccines, although the difference was only statistically significant when compared to the AMS3a E2E1-NP group ($p = 0.045$) and not when compared to mosaic E2E1-NP ($p = 0.15$). However, when we compared the normalized HVR1 binding titers, we found that the cocktail E2E1-NP group induced a more HVR1-focused response than mosaic E2E1-NP group ($p = 0.0035$; Fig. 5b). These data suggest that co-display of diverse E2E1 trimers on mosaic NPs can mitigate the immunodominance of HVR1.

Next, we measured neutralization against the six pseudoviruses that were matched with the cocktail and mosaic nanoparticle vaccine. The cocktail and mosaic group mounted the broadest NAb responses, while the NAb responses induced by the monovalent AMS3a E2E1-NP

group were skewed towards the autologous AMS3a virus (Fig. 5c). As expected, sera in the cocktail and mosaic E2E1-NP groups displayed broader neutralizing activity against the vaccine-matched viruses than AMS3a E2E1-NP (Fig. 5c, Table S1). We also measured neutralization against a panel of ten pseudoviruses from genotypes 1-4 that were mismatched to the mosaic and cocktail vaccines. Against this panel, the mosaic E2E1-NPs seemed to induce a slightly broader and more potent response than the cocktail of E2E1-NPs (Fig. 5d). In summary, the mosaic E2E1-NP elicited the most potent response when comparing the sixteen geometric mean titers (GMT) depicted in Fig. 5c, d : median GMT of 82 versus 31, $p = 0.0045$ for the comparison with AMS3a E2E1-NPs; 82 versus 65, not statistically significant, for the comparison with the E2E1-NP cocktail (Fig. 5e).

We also calculated the global geometric mean titer (GGMT) of each rabbit serum against the six vaccine-matched viruses and the ten mismatched viruses (Fig. 5f, Table S1). The GGMT of the vaccine-matched viruses was ~6-fold higher in the cocktail-immunized rabbits than those that received AMS3a E2E1-NP (median GGMT 268 versus 42, $p = 0.0039$), and ~2-fold higher in the cocktail group compared to the mosaic group, although the difference was not statistically significant (median GGMT of 268 versus 146, $p = 0.06$) (Fig. 5f). The GGMT of the mosaic immunogen against the mismatched viruses were significantly higher than that of the single E2E1-NP (median GGMT of 83 versus 34, $p = 0.014$ for the comparison). The difference with the cocktail nanoparticle vaccine formulation was smaller and not statistically significant (median GGMT of 83 versus 50). When we compared the ratios of the GGMT for vaccine-matched viruses and the GGMT for mismatched viruses, the difference between mosaic E2E1-NPs and the E2E1-NP cocktail was statistically significant ($p = 0.026$), suggesting that the mosaic nanoparticles were more likely to induce cross-reactive responses and cocktails of nanoparticles are more prone to induce autologous NAbs (Fig. 5f). Comparing the breadth (number of viruses neutralized) and potency (GGMT) of all animal sera revealed that the cocktail and mosaic E2E1-NP induced superior responses compared to E2, E2E1-I53-50A or the single E2E1-NP (Fig. 5g).

## Discussion

We generated a novel recombinant HCV E2E1 glycoprotein nanoparticle vaccine that elicited increased NAb titers, in terms of both breadth and potency, compared to E2 and E2E1 immunogens. Because of differences in dosage, adjuvants, animal species, number of immunizations, assay differences, and complications stemming from background neutralization in sera from mice and guinea pigs[31,67], it is difficult to directly compare our immunogenicity results to previous studies. However, single HCV E1E2-based immunogens usually did not elicit more potent or broader neutralization than single recombinant E2 immunogens when compared directly in an immunogenicity study, and our results are mostly in line with these earlier findings (see below)[36–38]. In contrast, mixing E2 immunogens from different strains can increase NAb breadth[68], while nanoparticle presentation can enhance potency[22,25,29]. Here, we combined these strategies by mixing different recombinant E2E1 immunogens on nanoparticles, either as cocktail or mosaic. This resulted in a significant improvement in NAb

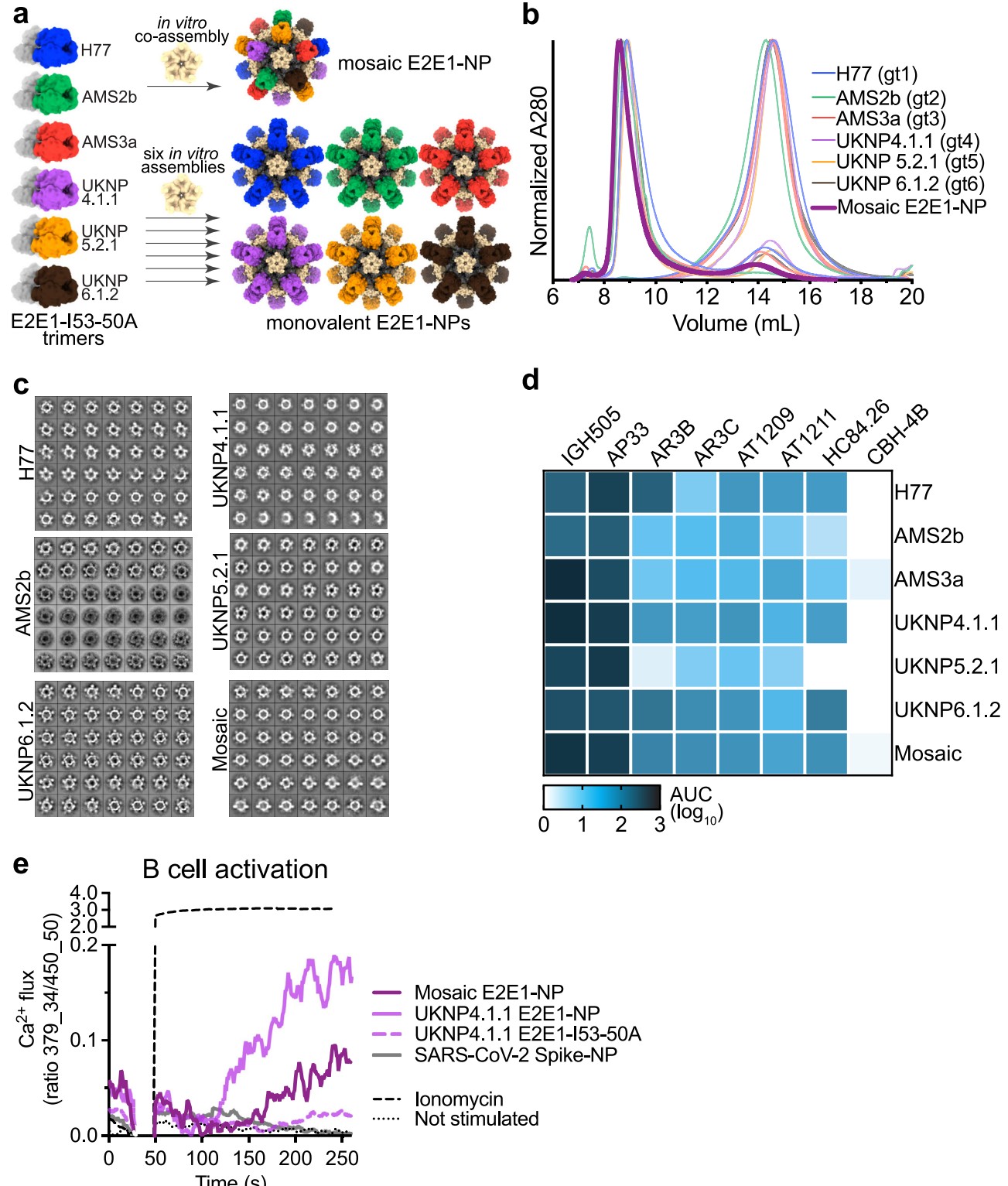

**Fig. 4 | Generation of mosaic E2E1 nanoparticles. a** Six E2E1-I53-50A trimers are combined and mixed with I53-50B pentamer to co-assemble as mosaic E2E1-NPs (top) or are individually mixed with I53-50B to assemble as monovalent E2E1-NP (bottom). **b** SEC profiles from a Superose 6 column of the mosaic E2E1-NP compared to six monovalent E2E1-NPs and the corresponding E2E1-I53-50A trimers. **c** NS-EM 2D class averages of mosaic and monovalent E2E1-NPs. **d** Binding of monovalent and mosaic E2E1-NPs to different mAbs measured by BLI. **e** B cell activation by E2E1-I53-50A trimers or E2E1-NPs. Activation was determined by measuring calcium flux in Ramos a B cell line harboring the AR3C bNAb. At 30 s, E2E1-I53-50A trimers or E2E1-NPs were added and calcium flux was measured after 20 s. SARS-CoV-2 spike nanoparticle (see ref. 48) and ionomycin were used as negative and positive control, respectively. Source data are provided as a Source data file.

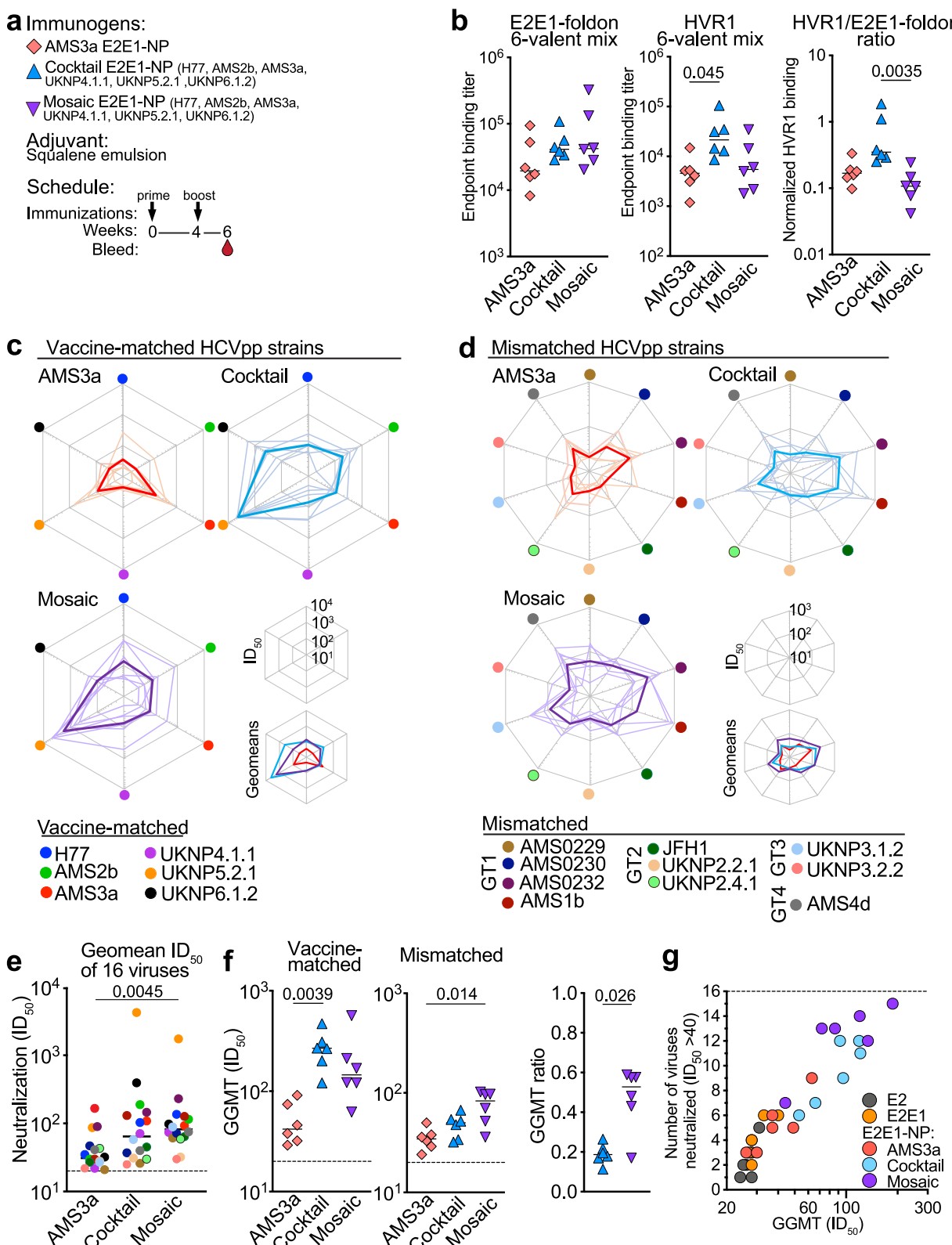

**a** Immunogens:
◇ AMS3a E2E1-NP
▲ Cocktail E2E1-NP (H77, AMS2b, AMS3a, UKNP4.1.1, UKNP5.2.1 ,UKNP6.1.2)
▼ Mosaic E2E1-NP (H77, AMS2b, AMS3a, UKNP4.1.1, UKNP5.2.1, UKNP6.1.2)

Adjuvant:
Squalene emulsion

Schedule:
Immunizations:
Weeks: 0 — 4 — 6

**b** E2E1-foldon 6-valent mix; HVR1 6-valent mix; HVR1/E2E1-foldon ratio

**c** Vaccine-matched HCVpp strains — AMS3a, Cocktail, Mosaic

**d** Mismatched HCVpp strains — AMS3a, Cocktail, Mosaic

Vaccine-matched:
● H77   ● UKNP4.1.1
● AMS2b   ● UKNP5.2.1
● AMS3a   ● UKNP6.1.2

Mismatched:
GT1: ● AMS0229  ● AMS0230  ● AMS0232  ● AMS1b
GT2: ● JFH1  ● UKNP2.2.1  ● UKNP2.4.1
GT3: ● UKNP3.1.2  ● UKNP3.2.2
GT4: ● AMS4d

**e** Geomean ID$_{50}$ of 16 viruses

**f** Vaccine-matched; Mismatched

**g**

E2
E2E1
E2E1-NP:
● AMS3a
● Cocktail
● Mosaic

---

breadth and potency compared to E2 alone, with up to 80-fold increase in potency, depending on the strain. The resulting NAb ID$_{50}$ titers are in the range of those associated with early clearance and/or protection against chronic infection in humans (1:80–1:100)[14,15]. However, neutralization titers obtained in rabbits might not be directly comparable with those that can be expected in humans. Higher titers and more durable responses might be needed to provide protection

from HCV infection in the real world. While improvements in antigen design (see below) might contribute to this, other factors such as better adjuvants, different routes of administration, masking of the nanoparticle scaffold, and additional booster immunizations, should also be considered[69–72].

Others have also presented designs that include permutating E1 and E2[36,37]. These studies describe a design in which E2 and E1 are

**Fig. 5 | Immunogenicity of cocktail and mosaic E2E1-NP vaccines. a** Rabbits were immunized at weeks 0 and 4 with AMS3a E2E1-NP (same group as Fig. 3), a cocktail of six monovalent E2E1-NPs or mosaic E2E1-NP. Antibody responses were measured at week 6. **b** Binding titers against a hexavalent mix of E2E1-foldon trimers (left) or hexavalent mix of HVR1 peptides (middle). Both mixes are sequence-matched to the cocktail and mosaic vaccines. Right: ratio between the HVR1 titer and E2E1-foldon titers. **c** Spider plots of the individual rabbit $ID_{50}$ titers (thin lines) and their geometric mean titers (GMT) (bold lines) against vaccine-matched HCVpp. **d** Spider plots of the individual rabbit $ID_{50}$ titers (thin lines) and the GMT (bold lines) against mismatched HCVpp. **e** Comparison of the GMT values from (**c**) and (**d**). Groups were compared using Friedman test and Dunn's post-test. **f** The global geometric mean titer (GGMT) against the six vaccine-matched (left) or ten mismatched

HCVpp (middle). Ratio of the GGMT against the mismatched and vaccine-matched HCVpps (right). Each dot represents a single rabbit. **g** Comparison of the NAb responses induced by AMS3a E2, E2E1-I53-50A and E2E1-NP (from Fig. 3) with those induced by cocktail E2E1-NP and mosaic E2E1-NP (Fig. 5). Each dot represents a single rabbit serum. GGMT was calculated over sixteen HCVpp strains. Breadth was defined as the number of viruses neutralized with an $ID_{50}$ titer >40. Horizontal lines in (**b**), (**e**), (**f**) indicate the median values. Significant differences between groups in (**b**), (**e**), (**f**) were determined using a Kruskal–Wallis test followed by Dunn's post-test, p-values are indicated; n = 6 rabbit sera per group (**b** and **f**) or n = 16 geomean $ID_{50}$ titers from 6 rabbit sera per group (**e**). P-values for significant differences are indicated on top of the graphs. Source data are provided as a Source data file.

directly fused or are separated by a covalent linker, while our design included the introduction of a furin cleavage site to allow non-covalent association of E1 and E2. Furthermore, the Guest and Prentoe designs included the hydrophobic regions at the C-termini of E1 (residues 326–352) and E2 (699–717), which might explain why these constructs mostly produce as higher molecular weight aggregates (~700 kDa)[37]. Importantly, these constructs also did not bind AR4A nor showed improved immunogenicity compared to E2[37].

In contrast to most other recombinant E1E2 designs, the E2E1-I53-50A immunogen represents a trimer of heterodimers instead of a single heterodimer. Historically, HCV E1E2 was assumed to be a class II fusion heterodimeric protein similar to flavivirus glycoproteins and membrane-associated E1E2 extracted from mammalian cells also forms a heterodimer[17,73]. However, structures of E2 and E1E2 did not reveal a typical class II fusion protein fold[17,20,21] and other studies have shown that E1E2 actually forms trimers of heterodimers on the virion surface[26,27]. Whether trimers and other oligomeric forms of E1E2 present functionally different conformations will require further functional and structural studies on virion-associated E1E2.

Nonetheless, E2E1-I53-50A has a number of potentially advantageous properties compared to current E2 immunogens. First, it conceals non-NAb epitopes, such as those of the CBH-4B, CBH-4D, and CBH-4G mAbs, probably because the trimeric conformation blocks access to these undesired epitopes. Second, we found that two PNGS (N623 and N645), are underoccupied on E2, probably resulting in unwanted glycan holes[74]. This might be relevant for immunogenicity, since holes in the glycan shield of HIV-1 envelope glycoprotein immunogens caused by underoccupancy of PNGS can attract undesired non-neutralizing Ab responses[74], while the absence of PNGS usually attract Abs that only neutralize the autologous strain[74–77]. In contrast, the N623 and N645 sites are fully occupied on E2E1-I53-50A, providing the latter with a denser glycan shield. Third, the increased valency of E2E1-I53-50A might have increase B cell activation and therefore showed slightly enhanced immunogenicity compared to monomeric E2 (Fig. S6). Other factors might contributed to these increased (N)Ab titers as well, not least the inclusion of the E1 domain. However, we acknowledge that the ~3-fold increase in NAb titers is lower than the ~13-fold increase in binding titers for E2E1-I53-50A versus E2 (Fig. S6). This suggests that the improved immunogenicity of E2E1-I53-50A trimers might be due to an increase in the quantity of Abs rather than an improvement in Ab quality.

We also realize that our current E2E1-I53-50A design probably does not represent the native conformation of viral E1E2, because it does not bind certain conformation-dependent bNAbs, such as AR4A and AT1618[4,13]. This property is shared with most recombinant E1E2 proteins that lack the native transmembrane domains[37,78,79]. However, a recent novel design in which the transmembrane domains were replaced by heterologous dimerization domains represents an important exception[36]. Other advances have been realized by rational optimization of bNAb epitopes, resulting in increases in the breadth and potency of NAb responses elicited after immunization[23]. Incorporating these aforementioned advances should facilitate the

improvement of the current E2E1-I53-50A design. Furthermore, the recent structure of membrane-associated E1E2 will aid the design of faithful mimics of the native E1E2 glycoprotein and capitalize fully on the progress made in structure-based immunogen design for other viruses[17,80–87]. For example, using structure-based design it might be possible to improve E2E1-I53-50A by optimizing the cleavage site between E2 and E1 and by shortening the linker between the E2E1 and I53-50A moieties,

The E2E1-NP immunogens elicited an immediate and strong Ab response against the HVR1 on E2. These anti-HVR1 Abs usually only target the autologous virus, but do not contribute to developing NAb breadth. The HVR1 shields conserved epitopes and probably stabilizes virion-associated E1E2[88]. Indeed, removing the HVR1 increases the neutralization sensitivity of virions[89–91]. However, simply removing the HVR1 from immunogens usually impairs their immunogenicity[37,92].

In contrast, E2 core antigens lacking the HVR1 have shown promising results in immunogenicity studies and are alluring antigens for epitope-focused vaccine strategies[22,24]. Furthermore, the most potent neutralizing sera in our study target two nearly overlapping epitopes (AR3/domain B and domain E) that are also present on E2 core (Fig. 3f). However, emerging evidence suggests that a vaccine eliciting antibodies against multiple non-overlapping antigenic regions is probably preferred over a vaccine eliciting NAbs against a single antigen region[13,93,94]. Furthermore, minimal antigens can induce antibodies against a neutralizing epitope but with an angle of approach that is incompatible with neutralization of the virus[95].

Therefore, we produced a vaccine in which we maintained the HVR1, in order to elicit NAbs that can cope with the restrictions imposed by the HVR1. By combining different immunogens on the same mosaic nanoparticle, we then tried to mitigate the anti-HVR1 response. Indeed, mosaic nanoparticles induced less unwanted anti-HVR1 responses, while eliciting cross-reactive NAbs more efficiently.

It is difficult to determine the exact stoichiometry of the different E2E1 antigens in the mosaic nanoparticles. In the context of corona vaccine research, we performed mass spectrometry measurements to determine the stoichiometry on I53-50-based mosaic nanoparticle displaying two coronavirus Spike proteins, those of SARS-CoV and SARS-CoV-2 (Brinkkemper et al. *submitted*). These experiments indicated stochastic assembly resulting in nanoparticles with a normal distribution of both Spike proteins. The resulting nanoparticle heterogeneity might pose a challenge for GMP manufacturing and quality control prior to clinical studies. However, the promising immunogenicity results presented in the current study and those obtained with mosaic nanoparticle vaccines against influenza and SARS-CoV-2[65,66,96] should provide an impetus to provide a regulatory framework for such heterogeneous nanoparticles. Furthermore, efforts are ongoing to design nanoparticles that assemble hierarchically, allowing for the production of homogeneous mosaic nanoparticles.

Nanoparticle presentation of antigens is a general strategy to improve antiviral antibody responses[47,49,97–100]. However, the success depends on whether the relevant NAb epitopes are properly displayed on the nanoparticle or are actually obscured by nanoparticle

presentation[47,97,101,102]. E2E1-NP mounted more potent and slightly broader Ab responses that competed with the binding of several E2-targeting bNAbs. This demonstrates that the geometry of E2E1 on I53-50NPs suitably presents these E2-specific cross-neutralizing epitopes. The I53-50NP platform we used here is being exploited for an RSV vaccine that is currently being evaluated in a phase I clinical trial[49,50]. Furthermore, an I53-50NP-based COVID-19 vaccine was recently approved in South Korea after successful completion of phase 1 and phase 3 studies providing a clear translational path forward for an HCV vaccine based on this platform[103].

In contrast to HIV-1 bNAbs, most HCV bNAbs have undergone relatively little somatic hypermutation[9,12,16], suggesting that neutralization breadth can be achieved early after antigen exposure, and that inducing them by vaccination should be feasible. Our demonstration that a cocktail or mosaic E2E1-NP vaccine can generate cross-reactive NAb responses after a two-dose vaccination regimen, further supports this notion and should pave the way for an effective HCV vaccine.

## Methods

### Constructs

All amino acids are numbered according to the standard H77 polyprotein numbering (GenBank AF009606[104]). Constructs were based on the HCV glycoprotein sequences from the following strains (GenBank ID; mutation(s)): AMS0232 (OL855837.1; I442F), H77 (AAB67037; R564C, V566A, G650E), UKNP2.1.1 (KU285220.1), AMS2b (KR094963.1), AMS3a (KR094964.1), UKNP4.1.1 (ALV85530.1), UKNP5.2.1 (ALV85536.1), UKNP6.1.2 (ALV85538.1). All recombinant E2 designs in this study, including when used as subunit of E1E2, E2E1, or E2E1-I53-50A, include the N-terminal HVR1 (384–411) and were truncated at position 698 at the C-terminus. Recombinant E2 (384–698) was fused to a GlySer-rich linker (GSGSGGRSG) and StrepII-tag (WSHPQFEK). For the E1E2 construct, the E1 ectodomain (192–325) was fused to the N-terminus of E2 (384–698), with the two subunits separated by a linker and optimized furin cleavage site (R6[43]). For E2E1, E2 (384–698) was fused to the N-terminus of the E1 ectodomain (192–325) separated by linker and R6 furin site. For E2E1-I53-50A, the StrepII-tagged I53-50A moiety (I53-50A1.NT1 from ref. 46) was fused to the C-terminus of E2E1, separated by a GS-4xGGS-linker. All constructs were codon-optimized for mammalian cell expression and were preceded by the same optimized tissue plasminogen (TPA) signal sequence to induce protein secretion. Amino acid sequences of the different constructs are listed in Fig. S2. Antibody sequences were human codon-optimized and cloned into human IgG1 expression vectors for the corresponding HCs or LCs as described before (Genscript Biotech)[105].

### HCV glycoprotein and antibody expression and purification

Suspension 293 F cells (Invitrogen, cat no. R79009) were maintained in FreeStyle medium (Life Technologies). Cells were transfected using 1 mg/mL PEI MAX (Polysciences Europe GmBH, Eppelheim, Germany) and DNA plasmid in a 3:1 (w:w) ratio at a density of 0.8–1.2 million cells/mL. E1E2, E2E1 or E2E1-I5350A were co-expressed with a plasmid expressing furin in a 1:1 ratio to ensure E1/E2 cleavage. The supernatant was harvested 6 days after transfection, centrifuged, and filtered using Steritops (0.22 μm pore size; Millipore, Amsterdam, The Netherlands) before further use. BioLock (IBA Life Sciences) and 1:10 (v:v) 10x Buffer W (1.0 M Tris-HCl, 1.5 M NaCl, 10 mM EDTA, pH 8.0) was added to the filtered 293F supernatant and flowed (0.5–1.0 mL/min) over a StrepTactinXT (IBA life sciences) column at 4 °C. The proteins were eluted with Buffer BXT (IBA life sciences) and concentrated in 10-kDa cut-off Vivaspin6 filters (Sartorius) in PBS (E2, E1E2, E2E1) or TBS/5% glycerol (E2E1-I53-50A). Subsequently, the proteins were run over a Superdex 200 Increase 10/300 GL or Superose 6 Increase 10/300 GL (GE Healthcare). His-tagged E2E1-foldon trimers were purified from 293 F supernatant using NiNTA agarose beads (Qiagen) and polished using a Superdex 200 Increase column in PBS. Antibodies were produced by co-transfecting the plasmids expressing IgG1 heavy chain (HC) and light chain (LC) in 293 F cells in a 1:1 ratio using PEI MAX. The 293F supernatants were harvested 5 days post-transfection, centrifuged, and filtered using 0.22 μm Steritop filters. The filtered supernatant was run over a protein A/G column (Pierce), followed by extensive washing with PBS and the antibodies were eluted with 18 mL 0.1 M glycine pH 2.5 directly captured in 2 mL neutralization buffer (1 M TRIS pH 8.7). The purified antibodies were buffer-exchanged to PBS using 100 kDa VivaSpin20 columns (Sartorius). Protein concentrations were determined with Nanodrop (Thermo Scientific, Wilmington DE, USA) using the molecular weight and extinction coefficient of the peptidic components only.

### I53-50B.4PT1 expression and purification

Lemo21 (DE3) (NEB) cells expressing I53-50B.4PT1 were grown in a 10 L BioFlo 320 Fermenter (Eppendorf) or a 2 L shake flask. Cells were grown in LB (10 g Tryptone, 5 g Yeast Extract, 10 g NaCl) at 37 °C to an OD600 of 0.8. After the cells were induced with 1 mM of IPTG, temperature was reduced to 18 °C and cells were grown for 16 h. The cells were then lysed in 50 mM Tris, 500 mM NaCl, 30 mM imidazole, 1 mM PMSF, 0.75% CHAPS using a Microfluidics M110P at 18,000 psi. Lysate was centrifuged at $24,000 \times g$ for 30 min. Clarified lysate was next applied to a Ni Sepharose 6 FF column (Cytiva) linked to an AKTA Avant150 FPLC system for immobilized metal affinity chromatography. I53-50B.4PT1 was eluted using a 30–500 mM imidazole linear gradient in 50 mM Tris pH 8, 500 mM NaCl and 0.75% CHAPS. Fractions containing I53-50B.4PT1 were pooled, concentrated using centrifugal filters with a 10,000 kDa cutoff (Millipore), sterilized and applied to a Superdex 200 Increase 10/300 (Cytiva) for further purification. Batches were tested to ensure low levels of endotoxin before use.

### E2E1-NP assembly

E2E1-I53-50 nanoparticles (E2E1-NP) were assembled essentially as described[47] with some adjustments. To remove aggregated protein, E2E1-I53-50A fusion proteins were passed through a Superose 6 Increase SEC column (GE Healthcare) in TBS/5% glycerol, pH 7.5. The column fractions containing non-aggregated E2E1-I53-50A trimers were pooled and mixed in an equimolar ratio with I53-50B.4PT1 (produced as described above) for an overnight (-16 h) incubation at 4 °C. The assembly mix was then concentrated at $1000 \times g$ using Vivaspin filters with a 10 kDa molecular weight cutoff (Sartorius) and passed through a Superose 6 Increase column in TBS/5% glycerol, pH 7.5. The fractions corresponding to the assembled NPs were pooled and concentrated at $500 \times g$ using Vivaspin filters with a 10 kDa molecular weight cutoff. Nanoparticle concentrations were determined with a Nanodrop using the peptidic molecular weight and extinction coefficient.

### Glycan analysis by HILIC-UPLC

Enzymatically released N-linked glycans were fluorescently labeled with procainamide using a 100 μL aliquot of labeling mixture (110 mg/mL of procainamide and 60 mg/mL of sodium cyanoborohydrate in 70% DMSO and 30% glacial acetic acid). This was added to the samples and incubated for 65 °C for 4 h. Labeled glycans were purified using Spe-ed Amide 2 columns (Applied Separations), as previously described[106]. Purified labeled glycans were analyzed using a 2.1 mm × 10 mm Acquity BEH Glycan column (Waters) on a Waters Acquity H-Class UPLC instrument with wavelengths $\lambda_{ex} = 310$ nm and $\lambda_{em} = 370$ nm. Endo H digestions of labeled glycans were used to measure abundance of oligomannose-type glycans.

### Site-specific glycan analysis

E2E1-I53-50A and E2 were denatured for 1 h in 50 mM Tris/HCl, pH 8.0 containing 6 M of urea and 5 mM dithiothreitol (DTT). Next, the

samples were reduced and alkylated by adding 20 mM iodoacetamide (IAA) and incubated for 1 h in the dark, followed by a 1 h incubation with 20 mM DTT to eliminate residual IAA. The alkylated proteins were buffer-exchanged into 50 mM Tris/HCl, pH 8.0 using Vivaspin columns (3 kDa) and two of the aliquots were digested separately overnight using chymotrypsin (Mass Spectrometry Grade, Promega) or alpha lytic protease (Sigma-Aldrich) at a ratio of 1:30 (w/w). The next day, the peptides were dried and extracted using C18 Zip-tip (MerckMilipore). The peptides were dried again, re-suspended in 0.1% formic acid, and analyzed by nanoLC-ESI MS with an Ultimate 3000 HPLC (Thermo Fisher Scientific) system coupled to an Orbitrap Eclipse mass spectrometer (Thermo Fisher Scientific) using stepped higher energy collision-induced dissociation (HCD) fragmentation. Peptides were separated using an EasySpray PepMap RSLC C18 column (75 μm × 75 cm). A trapping column (PepMap 100 C18 3 μM, 75 μM × 2 cm) was used in line with the LC prior to separation with the analytical column. The LC conditions were as follows: 280 min linear gradient consisting of 4–32% acetonitrile in 0.1% formic acid over 260 min followed by 20 min of alternating 76% acetonitrile in 0.1% formic acid and 4% ACN in 0.1% formic acid, used to ensure all the sample had eluted from the column. The flow rate was set to 300 nL/min. The spray voltage was set to 2.5 kV and the temperature of the heated capillary was set to 40 °C. The ion transfer tube temperature was set to 275 °C. The scan range was 375–1500 $m/z$. Stepped HCD collision energy was set to 15, 25, and 45%, and the MS2 for each energy was combined. Precursor and fragment detection were performed using an Orbitrap at a resolution MS1 = 120,000. MS2 = 30,000. The AGC target for MS1 was set to standard and injection time set to auto which involves the system setting the two parameters to maximize sensitivity while maintaining cycle time. Full LC and MS methodology can be extracted from the appropriate Raw file using XCalibur FreeStyle software or upon request.

Glycopeptide fragmentation data were extracted from the raw file using Byos (Version 4.0; Protein Metrics Inc.). The glycopeptide fragmentation data were evaluated manually for each glycopeptide; the peptide was scored as true-positive when the correct b and y fragment ions were observed along with oxonium ions corresponding to the glycan identified. The MS data was searched using the Protein Metrics 305 N-glycan library with sulfated glycans added manually. To search for N-glycans at non-canonical motifs, the above library was instead added as a custom modification to search for glycans (common1) at any N. The relative amounts of each glycan at each site as well as the unoccupied proportion were determined by comparing the extracted chromatographic areas for different glycotypes with an identical peptide sequence. All charge states for a single glycopeptide were summed. The precursor mass tolerance was set at 4 ppm and 10 ppm for fragments. A 1% false discovery rate (FDR) was applied. The relative amounts of each glycan at each site as well as the unoccupied proportion were determined by comparing the extracted ion chromatographic areas for different glycopeptides with an identical peptide sequence.

Glycan compositions were grouped according to their level of processing. HexNAc(2)Hex(10–5) compositions were classified as oligomannose-type, HexNAc(3)Hex(5-6)/HexNac(3)Hex(5-6)Fuc were classified as hybrid, and remaining glycan compositions were classified as complex-type.

## Enzyme-linked immunosorbent assay (ELISA)
For comparing antigenicity, purified StrepII-tagged E2 (1.0 μg/mL in TBS) or E2E1-I53-50A (equimolar E2, i.e., 2.1 μg/mL in TBS) were coated for 2 h at room temperature on 96-well Strep-TactinXT coated microplates (IBA LifeSciences). To determine serum binding titers, StrepII-tagged E2 (0.5 μg/ml) was coated on 96-well Strep-TactinXT plates (IBA LifeSciences) or (a mix of) His6-tagged E2E1-foldon trimers (1.0 μg/mL) was coated on 96-well Ni-NTA HisSorb (Qiagen) plates. Plates were washed with TBS twice before incubating with serially

diluted mAbs or CD81-LEL-hFc (R&D systems) in casein blocking buffer (Thermo Fisher Scientific) for 90 min. Rabbit sera were diluted in TBS/2% skimmed milk/20% sheep serum. After three washes with TBS, a 1:3000 dilution of HRP-labeled goat anti-human IgG (Jackson Immunoresearch) in casein blocking buffer or HRP-labeled goat anti-rabbit IgG (Jackson Immunoresearch) in 2% skimmed milk was added for 45 min. After washing the plates five times with TBS + 0.05% Tween-20, plates were developed by adding develop solution (1% 3,3′,5,5′-tetramethylbenzidine (TMB, Sigma-Aldrich), 0.01 % $H_2O_2$, 100 mM sodium acetate, 100 mM citric acid) and the reaction was stopped after 3 min by adding 0.8 M $H_2SO_4$. Absorbance was measured at 450 nm. Binding endpoint titers were set at 5x cutoff and determined in Graphad Prism 8.3.

## Competition ELISA
HCV bNAbs were biotinylated using the EZ-Link biotinylation kit (ThermoFisher, cat# 21335) according to the manufacturer's instructions. HEK293T cells in a T-75 flask (~70–80% confluency) were transfected with AMS3a E1E2 plasmid using PEI MAX. After 72 h, the cells were harvested and lysed with 2.0 mL of 1.0% Triton X-100 in TBS (~10 × 10⁶ cells/mL). After 30 min, the lysate was clarified by centrifugation (1500 × g), filtered (0.22 μm filter), aliquoted and the E1E2-containing lysates were stored at −80 °C. Half-well 96-well plates were coated with *Galanthus nivalis* lectin (Vector Laboratories) at 20 μg/mL in 0.1 M NaHCO₃ pH 8.6. The next day, the plates were blocked with casein blocking buffer and then E1E2 lysate (1:100 diluted in TBS) were added to the lectin-coated plates. After 2 h, the plates were washed (TBS) and rabbit sera (1:50 dilution in casein). After 30 min, biotinylated HCV bNAbs were then added at a concentration to give 50–80% of maximum binding signal without competitor (determined in a pilot experiment) and incubated for 1 h together with the rabbit sera. Plates were washed (TBS) and bound biotinylated HCV bNAb was detected using streptavidin-polyHRP40 (Fitzgerald, #651R-S104RHRP) (1:10,000 in casein). Plates were developed as described above for ELISA. The remaining binding signal of each well was subtracted from the negative control that contained serum but no bNAb (set to 0%) and compared to binding signal of the wells with bNAb only (set to 100%).

## Bio-layer interferometry (BLI)
Antibody binding to purified proteins was measured using a ForteBio Octet K2. All assays were performed at 30 °C and with agitation set at 1000 rpm. E2, E2E1-I53-50A, and E2E1-NPs and antibody dilutions were made in running buffer (PBS/0.01% BSA/0.002% Tween-20) in a final volume of 240 μL/well. Antibody was loaded on protein A sensors (ForteBio) at 3.0 μg/mL in running buffer until binding threshold of 1.0 nm was reached followed by a baseline of 30 s. Proteins were diluted in running buffer at 100 nM (equimolar E2 monomer) and association and dissociation were measured for 300 s. For background correction, binding signal of running buffer to protein A sensor with antibody was measured. Area under the curve was calculated by first subtracting the background signal and calculated using Graphpad Prism 8.3.

## Negative-stain EM
Negative-stain EM experiments were performed as described previously[47]. E2E1-I53-50A and E2E1-NP samples were diluted to 20–50 μg/mL and loaded onto the carbon-coated 400-mesh Cu grid that had previously been glow-discharged at 15 mA for 25 s. Grids were negatively stained with 2% (w/v) uranyl formate for 60 s. Data collection was performed on a Tecnai Spirit electron microscope operating at 120 keV. The magnification was ×52,000 with a pixel size of 2.06 Å at the specimen plane. All imaging was performed with a defocus value of −1.50 μm. The micrographs were recorded on a FEI Eagle CCD (4k) camera using Leginon automated imaging interface. Data processing was performed in Appion data processing suite. With nanoparticle

samples, ~500–1000 particles were manually picked from the micrographs and 2D-classified using the Iterative multivariate statistical analysis (MSA)/multireference alignment (MRA) algorithm. With E2E1-I53-50A trimer samples, 10,000–40,000 particles were auto-picked and 2D-classified using the Iterative MSA/MRA algorithm.

## B cell activation assay
AR3C-carrying B cells were generated, essentially as described elsewhere[48]. In short, the gl2-1261 gene of the pRRL EuB29 gl2-1261 IgG TM.BCR.GFP.WPRE plasmid[107] was exchanged for the heavy and light chain genes of AR3C[5] using Gibson assembly (Integrated DNA Technologies). Lentiviruses were produced by co-transfecting the expression plasmid with pMDL, pVSV-g, and pRSV-Rev into HEK293T cells using lipofectamine 2000 (Invitrogen). Two days post transfection, IgM-negative Ramos B cells were transduced with HEK293T supernatant. Seven days post transduction, AR3C-expressing B cells were FACS sorted on IgG and GFP double-positivity using a FACS Aria-II SORP (BD Biosciences) (see Fig. S11 for the gating strategy). B cells were expanded and cultured indefinitely. B cell activation experiments of AR3C-carrying Ramos B cells were performed as previously described[48]. In short, 4 million cells/mL in RPMI++ were loaded with 1.5 μM of the calcium indicator Indo-1 (Invitrogen) for 30 min at 37 °C, washed with Hank's Balance Salt Solution supplemented with 2 mM $CaCl_2$, followed by another incubation of 30 min at 37 °C. Antigen-induced $Ca^{2+}$ influx of AR3C B cells were monitored on a LSR Fortessa (BD Biosciences) by measuring the 379/450 nm emission ratio of Indo-1 fluorescence upon UV excitation. Following 30 s of baseline measurement, aliquots of 1 million cells/mL were then stimulated for 210 s at RT with either 20 μg/mL of UKNP4.1.1 E2E1-I53-50A trimers, 25 μg/mL of UKNP4.1.1 E2E1-NP or mosaic E2E1-NP (i.e., equimolar trimer). Ionomycin (Invitrogen) was added to a final concentration of 1 mg/mL to determine the maximum Indo-1-fluorescence. 20 μg/mL of SARS-CoV-2 spike nanoparticle was used as negative control[48]. Kinetics analyses were performed using FlowJo v8.1.

## Rabbit immunizations
In a first immunization study, 18 rabbits (New Zealand White, female, 3 groups, 6 animals/group) were immunized under subcontract at Covance (Denver, USA) with either 10 μg AMS0232 E2 monomer, 22 μg AMS0232 E2E1-I53-50A or 27 μg AMS0232 E2E1-NP (equimolar to E2). Antigens were mixed 1:1 with squalene o/w emulsion (SE) (i.e., 250 μL of antigen in PBS combined with 250 μL SE) (Polymun, Klosterneuburg, Austria) and administered by two intramuscular immunizations in each quadriceps (2 × 250 μL). Rabbits were bled at weeks 0, 4, and 6. All immunization procedures complied with all relevant ethical regulations and protocols of the Covance Institutional Animal Care and Use Committee (IACUC, study C0096-19). In a second study, 30 rabbits (New Zealand White, female, 6 animals/group) were immunized under subcontract at Pocono Rabbit Farm & Laboratory (Canadensis, USA). The rabbits received 10 μg AMS3a E2 monomer, 22 μg AMS3a E2E1-I53-50A or 27 μg AMS3a E2E1-NP, a cocktail of 27 μg E2E1-NPs (6 × 4.5 μg) or 27 μg of mosaic E2E1-NP. The antigens were formulated in SE adjuvant, the rabbits received two intramuscular immunizations in each quadriceps (2 × 250 μL) at weeks 0 and 4. Immunization procedures complied with the relevant ethical regulations and protocols of the Pocono Institutional Animal Care and Use Committee (IACUC).

## Neutralization assays
For generating the HCVpps, $1.5 × 10^6$ HEK293T cells were seeded on a 10 cm dish a day prior to transfection. We co-transfected E1E2, a MLV Gag-Pol packaging construct and firefly luciferase[108] in optimized ratios (1:1:1 or 1:2.6:3.4 or 1:26:34)[109] with a total amount of 6 μg of DNA and 12 μl of Lipofectamine 2000 (Invitrogen) in Opti-MEM (Thermo-Fisher). Opti-MEM was replaced by DMEM (Gibco)/10% fetal bovine serum (FCS)/0.1% Penicillin-Streptomycin (PS) the next day. Two days later, the supernatant containing the HCVpps was passed through a 0.45 μm filter and frozen at −80 °C for long-term storage or 4 °C when used within a week.

One day prior to the neutralization assay, $15 × 10^3$ Huh-7 cells per well in a 96-well plate were seeded in 100 μL DMEM/10% FCS/0.1% PS/1% nonessential amino acids/1% HEPES buffer (Huh-7 medium). HCVpps were incubated with sera in duplicate at 37 °C in 5% $CO_2$ which were serially 3-fold diluted starting with a 1:20 dilution. After 1 h, the HCVpp/serum mixture was added to the Huh-7 cells and incubated for 4 h. Next, 200 μL Huh-7 medium was added and incubated for 72 h. Cells were lysed and luciferase signal was measured using the SteadyGlo luciferase reagent kit and a GloMax luminometer (Promega, USA).

Neutralization data was analyzed with Graphpad Prism (version 8.3) and stored in Microsoft Excel (version 16.5).

## Reporting summary
Further information on research design is available in the Nature Portfolio Reporting Summary linked to this article.

## Data availability
Other data that support the findings of this study are available from the corresponding authors (K.S. and R.W.S.) upon reasonable request. Designed constructs are based on sequences deposited at GenBank: AMS0232 (OL855837.1; mutation I442F), H77 (AAB67037; mutations R564C, V566A, G650E), UKNP2.1.1 (KU285220.1), AMS2b (KR094963.1), AMS3a (KR094964.1), UKNP4.1.1 (ALV85530.1), UKNP5.2.1 (ALV85536.1), and UKNP6.1.2 (ALV85538.1). Source data are provided with this paper.

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

## Acknowledgements
We thank Mitch Brinkkemper, Marielle van Breemen, Jonne Snitselaar, and Tom Bijl for experimental support. We thank Steven Foung for donating the CBH-4B, CBH-4D, and CBH-4G antibodies and Tim Beaumont and Sabrina Merat for donating the AT1209 and AT1211 antibodies. We thank Dietmar Katinger and Philipp Mundsperger for providing the squalene emulsion adjuvant. The following reagent was obtained through the NIH AIDS Reagent Program, Division of AIDS, NIAID, NIH: Ramos B cells from Drs. Li Wu and Vineet N. Kewal Raman. We thank Andrew McGuire for kindly sharing the pRRL.EuB29 lentiviral vector that was used to transduce Ramos B cells. This research was supported by the Fondation Dormeur, Vaduz (to R.W.S. and to M.J.v.G.), an AMC Fellowship from Amsterdam UMC (M.J.v.G.), Vici grant from the Netherlands Organization for Scientific Research (NWO) (R.W.S), a Vidi and Aspasia grant from the NWO (grant numbers 91719372 and 015.015.042) (J.S.), the Bill & Melinda Gates Foundation (OPP1156262 to N.P.K. and R.W.S.), an Amsterdam institute for Infection and Immunity Postdoctoral grant (K.S.), and an AMC PhD Scholarship (A.C.M.). Mass spectrometry and electron microscopy (M.C. and A.B.W.) were supported by Bill and Melinda Gates Foundation grant INV-008352/OPP1153692. Y.W. has taken up a position at AstraZeneca; all experimental work was performed prior to this development. The funders had no role in study design, data collection and analysis, decision to publish, or preparation of the manuscript.

## Author contributions
Conceptualization: K.S. and R.W.S. Funding acquisition: K.S., J.S., and R.W.S. Investigation: K.S., L.R., J.C-P., Y.W., I.Z., A.C., W-H.L., M.d.G., J.K., S.K., and G.O. Methodology: K.S., I.Z., A.C., I.d.M-S., and P.J.M.B. Project administration: K.S. Resources: R.R. and N.P.K. Supervision: K.S., A.B.W., M.J.v.G., M.C., J.S., and R.W.S. Writing—original draft: K.S. and R.W.S. Writing—review & editing: all authors.

## Competing interests
N.P.K. is a co-founder and shareholder of Icosavax, a company that has licensed patent applications regarding the I53-50 nanoparticle system, and N.P.K. is a member of Icosavax's Scientific Advisory Board. All other authors declare no competing interests.
