## [Peer Review File · Nature Communications]

Induction of cross-neutralizing antibodies by a permuted hepatitis C virus glycoprotein nanoparticle vaccine candidateREVIEWER COMMENTS

Reviewer #1 (Remarks to the Author):

The authors describe the design of permuted E1E2, its fusion to self-assembling nanoparticles, and the use of this nanoparticle construct as a mosaic vaccine. While the testing of E1E2 antibody binding seems to indicate that the described permuted design does not represent a native-like E1E2 heterodimer, which is disappointing, the results presented regarding the use of the E1E2 design in a nanoparticle and mosaic format should be of interest to the research community. Several concerns, noted below, should be addressed by the authors.

1. It is not clear from the text why the authors refer to their E1E2 design as a “circular permutation” versus a “permutation”. As there are just two proteins (E1 and E2), E2E1 seems to be just a permutation (and the only possible permutation), and the “circular” aspect of the permutation has no meaning in this context. If the authors wish to refer to the design as “circular”, they should explain the meaning of that somewhere in the introduction or results where the design is first mentioned and referred to in that manner. Otherwise, they can just refer to the design as a permutation, to avoid possible confusion for readers.

2. The apparent lack of binding to AR4A (Fig. S1e) raises the question of whether the E2E1 assembly designed by the authors resembles native E1E2 or can present any of the known E1E2-dependent antibody epitopes. Even with lack of AR4A binding, other known E1E2 antibodies are available, such as AR5A, or the monoclonals described by Colbert et al. (J Virology 2019), but it is not clear whether the authors tested any antibody outside of AR4A to confirm the nature of the E1E2 assembly. Lack of binding to “certain conformation-dependent bNAbs” is noted on line 377 in the Discussion, suggesting that multiple E1E2 bNAbs were tested instead of a single antibody. Accordingly, the authors should test their design for binding to at least one more E1E2 antibody, such as AR5A, to help readers better understand whether or not the design reflects native E1E2-like antigenicity and can present E1E2-dependent epitopes.

3. On lines 136-137, the authors note the “inherent flexibility of the HCV E1E2 glycoprotein” as a possible cause for diffuse E2E1 EM density, however the authors have recently described the high resolution structure of the E1E2 complex (Torrents de la Peña BioRxiv 2021, cited in this manuscript), which indicates that E1E2 is not too flexible for obtaining high resolution structures, contradicting their statement in this manuscript. It seems possible that lack of stability, or added flexibility or dynamics, of the E2E1 design, may be responsible for the diffuse density, and the authors should consider noting this as a possible cause.

4. The lack of binding of the constructs to AR3C is concerning (Figure 1h) and should be addressed by the authors in the Results text. At very least, the E2 control protein would be expected to bind AR3C, and no E2 binding seems to suggest a systematic error with the assay (e.g. a poor-quality antibody), or some properties of the HCV strain’s (AMS0232) E2 and E1E2 sequences that do not permit AR3C binding.

5. The ID50 titers noted in Figure 2 and in the associated Results text seem quite low, and it is not clear whether the quantities noted (e.g. geometric mean titers of 34, 21, and an ID50 cutoff of 40) are reliable for comparisons, given the background that is sometimes observed in HCVpp assays. The authors should provide information in the Methods regarding the starting serum dilution level used for the neutralization assays (e.g. 1:20 or 1:40), and they should provide example curves from a neutralization assay of immunized and pre-immune rabbit sera (at least one rabbit per group) in a supplemental figure.

6. It is not clear why the authors selected the permuted version of E1E2 (E2E1) over the non-permuted version for their nanoparticle and in vivo studies, at least based on the antigenicity data

shown in Figure S1. Although the hypothetical proximity of the E2 C-terminus to the E1 N-terminus is noted by the authors as a rationale for the permuted design, it still seems theoretically possible that E1E2-I53-50A would be antigenic and would form nanoparticles. The authors should comment on why the selection of E2E1 was made for nanoparticle fusion, or show the data that were used to support that.

7. At least two previous studies have reported permuted E1E2 constructs (Prentoe et al. 2021, Guest et al. 2021, cited in this study). The authors do not seem to discuss these E2E1 precedents; they should be noted by the authors (along with E2E1 precedent studies, if applicable), and briefly compared with the currently described E2E1 design.

Reviewer #2 (Remarks to the Author):

The aim of this study was to examine the potential of nanoparticles comprised of the E1/E2 glycoproteins of HCV to elicit bNAb in vaccinated rabbits. In the past, this has proved to be a difficult as the virus shows high sequence variability in these envelope proteins which are the target of NAb.

The study was well conceived, the manuscript was well written and supported by an appropriate bibliography and comprehensive figures.

The study is the result of a considerable amount of work, not least the glycan shield studies and builds on a previous study of the E1/E2 structure of the virus by the authors to apply the 153-50NP technology used previously in COVID-19 and RSV vaccine studies to HCV.

Several points should be addressed;

Lines 39 et seq; As some current licensed vaccines use a cocktail and as a number of research groups are investigating the use of a cocktail for different infectious agents, I believe that the abstract should reflect the interesting data viz. that the mosaic was more effective than a cocktail in eliciting bNAb.

Lines 66 et seq; It is not clear if the authors equate an established infection with a persistent infection. In my view, and according to the general dogma, cell mediated immunity is generally required to clear an established (persistent) infection, and earlier studies from the Chisari group demonstrated this very well. As there is no evidence for ADCC as a potential mechanism for clearance of HCV-infected hepatocytes, the authors should dilute the impact of NAb on clearance and introduce a (important) role for cellular immunity in this process.

Lines 176 et seq; As I read the text and compared the text in this section with the figures, I noted that the authors failed to mention any comparison between the E2E1-NP and the E2E1-15350A trimers until line 208, essentially as a throw-away line. I kept asking why the authors failed to present this comparison, and as I'm certain that other readers will think similarly, I believe that this is an important comparison that should be given greater emphasis in the text from line 222.

Lines 280 et seq; what was the basis for choosing UKN4.1.2 NPs for the B cell activation studies?

Minor points;

Line 51; suggest "persist" for "linger"

Line 106; insert "the TMD and" immediately prior to "the predicted membrane-proximal"

Line 118; insert "the particles" immediately prior to "purified"

Figure legends; in figures which show the NS-EM images, the white bar is not mentioned in the legends until figure 4 which does not contain NS-EM images

Reviewer #3 (Remarks to the Author):

Review of manuscript NCOMMS-22-07463

Although direct-acting antiviral (DDAs) can cure most hepatitis C virus (HCV) infections, they cannot prevent liver damage and reinfection. Development of a broadly protective HCV vaccine thus remains a top priority. In this manuscript, Sliepen et al. presented an HCV vaccine strategy that displays soluble E1E2 dimers derived from one strain, or different genotypes (mosaic), on a two-component nanoparticle (I53-50) as vaccine candidates. The authors then immunized rabbits to compare the neutralizing antibody (NAb) responses induced by E2, E2E1-I53-50A trimer, and E2E1-NP. The results showed moderate improvement of breadth and potency. Overall, this is an interesting study carried out by a group of vaccine experts. However, despite the novelty of the vaccine design, there are a lot of concerning issues that may not be easily addressed. Some self-conflicting data or statements need to be clarified or corrected during revision.

Major comment #1:

There is not enough evidence to support the notion that HCV glycoprotein is a homotrimer of E1E2 heterodimer like HIV-1 Env, and HCV is not a class-I fusion virus. The Falson study only had low-resolution (SDS-PAGE) data, whereas the Freedman study was based on computational modeling only. The authors' statements on page 4, lines 79-83, seem to be self-contradictory. If they already solved a high-resolution structure of native E1E2 dimer on the cell membrane, how can this data be used to support E1E2 being a trimer on the virion surface? Are there any studies on the E1E2 multimeric state when displayed on cell versus virus surface?

Major comment #2:

On page 4, lines 107-108, and page 5, 109-112, the authors designed soluble E1E2 and E2E1 and selected E2E1 for further development based on the Alanine-scanning work of Giang et al. and Gopal et al. It is somewhat puzzling why the authors used such low-resolution data, and not their high-resolution cryo-EM structure, for this most critical decision in construct design. As the top experts in structure-based vaccine design, the authors need to provide some good explanations. The authors are also encouraged to confirm, based on their cryo-EM data, that the N terminus of E1 is indeed close to the C terminus of E2, which is the basis of their E2E1 design. Finally, the authors inserted the same 15-aa GS linker in both E1E2 and E2E1 constructs, contradicting their statement that they picked the E2E1 design based on the Ala-scanning data.

Major comment #3:

On page 5, lines 133-138, the authors first claimed a homogeneous E1E1-I53-50A trimer based on the NS-EM data and then said that they were unable to get a higher-resolution reconstruction due to flexibility. First, the same data cannot be used to support opposite conclusions depending on what the authors need. Second, since the authors have determined a high-resolution cryo-EM structure for the full-length, native E1E2 dimer, they should have more than enough expertise to get a reasonable NS-EM model. Third, if flexibility is indeed the problem, does that mean further optimization of the linker length is required at this point?

Major comment #4:

On page 5, lines 167-168 (Figure 1h), the authors measured the antigenicity of three constructs against a panel of NABs. However, AR4A, a well-known bNAb that only recognizes the native E1E2 dimer, was not included in the antigenic evaluation. It was then stated on page 13, lines 376-378, that "We realize that our current E2E1-I53-50A design probably does not represent the native conformation of viral E1E2, because it does not bind certain conformation-dependent bNABs, such as AR4A (Giang et al., 2012)". This is a fundamental problem of this study.

Over the last decade, the same group of authors emphasized the importance of native-like Env trimers in the HIV vaccine field and coined the terms “pseudotrimers” or TINOs (trimer-in-name-only) to describe those Env antigens that cannot bind conformation-dependent bNAbs such as PGT145. The lack of AR4A binding suggest that E1E1-I53-50A suffers a similar problem to those HIV pseudotrimers and TINOs. According to the authors’ many reviews in the HIV field, such non-native antigens should not be used in vaccine development.

Another issue is that the authors already reported a high-resolution cryo-EM structure for the native E1E2 dimer in complex with AR4A. It is puzzling why in this study they would move forward with a problematic antigen design when they could easily access the correct answer. The authors need to provide convincing explanations for all these choices.

Major comment #5:

On page 7, lines 180-181, the authors created an E2E1-foldon probe for analyzing E2E1-directed binding antibody responses in vaccination. The same structural and antigenic evaluation must be done for E2E1-foldon to ensure that the same E2E1 conformation is presented on I53-50A and foldon, otherwise, the response measured by this probe would be biased.

Furthermore, the use of an E2E1 probe to analyze rabbit serum will no doubt result in a much lower reading for the E2 group and favor the two vaccine antigens that contain both E1 and E2. The authors should include an E2 probe in the analysis, which could provide insight into: (1) the impact of multivalent display on antibody titer (since all three antigens contain E2), and (2) the effect of E1 on E2-directed antibody response, in both a trimeric form and an NP form.

Major comment #6:

On page 7, 193-197, the authors reported enhanced HVR1-specific antibody responses for E2E1-NP, raising the concern that the multivalently displayed HVR1 will distract immune system from more conserved epitopes such as the CD81-binding loop, front layer, and AS412 that are targeted by bNAbs. Although the mosaic E2E1-NP design was intended to reduce the HVR1-specific antibody response by diversifying its sequence, the concern is that a large repertoire of weak or non-neutralizing HVR1 antibodies would be generated by such mosaic vaccines.

Major comment #7:

On page 9, lines 246-249, the authors reported an interesting finding that the AS412 and AR3C epitopes contributed to the increased neutralizing breadth seen for the E2E1-NP group. This begs the question whether an E1E2 or E2E1 dimer-based vaccine strategy is truly superior to an E2-based vaccine strategy, because the AS412 and AR3C epitopes are on the E2 surface and remain intact in all E2 core designs after truncation of HVR1, VR2, and VR3. Based on this finding, the authors may want to discuss the different goals of the dimer and E2-based vaccine strategies.

Major comment #8:

On page 10, lines 273-275, the authors missed E2E1-I53-50A trimers of 6 genotypes with I53-50B to form mosaic NPs. Since a fully assembled I53-50 NP can present 20 trimers and there are 6 different E2E1-I53-50A trimers, the final mosaic E2E1-NP vaccine may be a mixture of many forms with different genotype ratios, structural layouts, and melting temperatures. How can such a mosaic vaccine be characterized and pass a rigorous QC system in GMP manufacturing? The authors need to discuss these potential caveats related to mosaic NP vaccines.

Major comment #9:

Discussion, page 12-14, needs be revised to include objective and balanced discussions about the issues listed in #1-8. For example, on page 14, lines 407-409, it is claimed that “the geometry of E2E1 on I53-50 NPs suitably presents the desired cross-neutralizing epitopes”. However, on the previous page 13, lines 376-378, there is a statement that “it doesn’t bind certain conformation-dependent bNAbs, such as AR4A”. These two statements contradict each other. The native-like E1E2

construct reported by Guest et al. is clearly more suitable for NP vaccine development.

Major comment #10:

On page 14, lines 409-412, the authors mentioned the COVID-19 vaccine based on the I53-50 NP platform as supporting evidence for their HCV E2E1-I53-50 NP vaccine. Some web search revealed that this COVID-19 vaccine (Walls et al., 2020) was licensed to a company, Icosavax, which conducted a phase-I/II trial with early data released on March 25. The results were "below the company's expectations" and "discordant with the preclinical data" published in Cell (Walls et al., 2020). In fact, the vaccine response appeared to be "comparable to or below the placebo". In some reports, it was suggested that the platform has a "fatal flaw". In light of this new human data on the I53-50 NP platform, the authors still need to discuss the potential implications and how this would affect the translational path for their HCV vaccine based on the same platform.

Reviewer #4 (Remarks to the Author):

The authors describe a novel two-component vaccine approach for making an HCV vaccine. The approach is interesting as it allows for easy incorporation of higher order forms of the HCV envelope proteins (here E2E1 trimers most likely), which cannot be easily accommodated in "standard" one-component VLP vaccine approaches such as previously described for HCV. Although the overall improvement using the mosaic E2E1-NPs is modest it appears that the conclusion that this two-component vaccine merits further study is sound. In addition, the authors have performed an impressive array of antigen/immunogen characterizations, which also has relevance for HCV vaccine development. This includes glycan analyses of purified protein variants and their extensive testing of mosaic or cocktail NPs, which unfortunately yielded very modest improvements compared with a single isolate E2E1-NP. Their data that E2E1 trimers represents a promising HCV vaccine antigen is less convincing. The initial comparison with what they refer to as "monomeric E2" does not show that E2E1 trimers are in fact superior. In fact, it seems the opposite may well be true (see below). They attempt to compare serum antibodies derived from mice immunized with lower order E2 complexes (probably not monomers, and certainly a mix of E2 species; see Figure S6b) with E2E1 (which form >400 KDa complexes, maybe trimers) and even larger E2E1-NPs. Not surprisingly, E2 does not induce the same levels of E2 or E1E2 antibodies, which is likely due to its smaller size. The authors themselves find that overall antibody levels correlate with neutralization (Figure 2f). Moreover, the authors report that anti-E2E1 antibodies are elevated 60 fold for the E2E1-NPs compared with E2, whereas neutralization is only increased 10-fold (l. 240; Results). This indicates that while the antibody levels are, unsurprisingly, increased on E2E1 and E2E1-NPs (partly due to size and increased avidity of epitopes), the quality of those antibodies is decreased (around 6 fold). Their epitope exposure/binding tests using Octet also shows that while E2E1 binds less to several non-neutralizing MAb's it also has reduced binding to AR3B, AT12009, AT12011 and HC84.26). In fact, the only E2 antibody that binds more strongly to E2E1 trimers is the AP33, which targets a linear epitope. Taken together this may indicate that E2E1 does not fold properly, which is also in line with the lack of AR4A binding, and likely folds "worse" than E2 alone (at least from a vaccine antigen perspective). A proper control would be to purify a single species of E2 (monomer, dimer or something else) and then attach this to their NP and test immunogenicity in comparison with E2E1-NPs. For these reasons I disagree with several of the author's key claims as presented both in the Results section as well as in the Discussion.

In addition, other major/minor points of interest/concern are listed below:

Major general issues:

Generally, I think the authors could spend a bit more time briefly describing how the methods they use allow them to address the questions they ask. As it is now a reader would have to be quite knowledgeable in a wide range of scientific methodology to easily read the manuscript (see also some of the minor specific issues for cases where this is particularly problematic).

Major specific issues:

It is not possible to see the E2 band in Figure S1d. Unless the size of E2 is obscured by the E1 band (which the authors do not mention) this is unexpected. It is also of some concern as it is possible that E2 is partially lost to the complex once furin cleavage occurs. A repeat with more protein, clearly showing both E1 and E2 preferably with WB data as well, for both proteins would be a very relevant and simple control to perform.

L. 131. Results. The authors do not see binding of AR4A to the potential E2E1 trimers. As the authors acknowledge in the Discussion, this likely indicates that E2E1 trimers are not in a native E1E2 conformation with resulting differences in binding for some MABs. It would be relevant to test binding of other MABs known to require an E1E2 interaction for binding, such as AR5A or HEPC146.

Minor general issues:

It seems an oversight on behalf of the authors not to mention the two Christiansen et al. papers on an HCV E2 VLP vaccine (2018; 2019, Sci Rep).

The use of the term "circular" permutation is not explained. I am guessing that it simply refers to the swapping of E1 and E2 in the primary coding sequence, but the authors may mean something different. It should be explained clearly.

The abbreviation HILIC—UPLC is used without spelling out the acronym in the main text. Also, I think it would be helpful to general readers with a line or two concerning what the technique accomplishes (i.e. how it enables glycan analysis).

The authors refer to BLI, but make no mention of it in the method section, leaving it up to the readers to know (or guess) that the Octet is in fact a BLI instrument. The BLI data is also shown only as relative intensities in a color table, making it difficult to assess how substantial the differences really are. A supplementary figure with the actual binding curves would be very nice.

The authors' NPs that are populated with several different E2E1 trimers (mosaic NPs). However, it does not seem like they have any measure of whether the different E2E1 trimers are bound to the NPs at the same frequency (i.e. the E2E1 stoichiometry on the particles). Is there some way to measure this? The authors claim that the mosaic E2E1 NPs induce less HVR1-specific antibodies, but that is not clear from the 6-valent HVR1 ELISAs they perform. It is good they normalize the HVR1-specific antibody responses against the overall antibody response. However, they do not find differences for mosaic E2E1-NPs compared with AMS3a-NPs, which is not surprising as antibodies against other HVR1 sequences (present in the mix) will obscure any reduction in AMS3a HVR1-specific antibodies. They should supplement this by running an AMS3a HVR1-specific ELISA for the same mouse sera as shown in Figure 5b.

Minor specific issues:

I could not find a specific reference to the sequence and truncation sites of soluble E2. The assumption would be that it is truncated at 698 and of the AMS3032 isolate, but this should be made clear both in the Methods and in the Results section, where the E1E2 and E2E1 constructs are explained.

L 52. Intro. I do not think it is clear, or even likely, that long term ill-effects of HCV infection is linked to hepatocyte killing. Long-term liver inflammation is believed to play a major role in this. As it is not central to the study itself, I recommend taking out the information regarding the way in which HCV leads to HCC and other liver-related disease.

L 93. Intro. "Sofar" as two words.

L. 94. Intro. Soluble E1E2 trimers should be E2E1 trimers.

L. 126. Results. The authors state that the E2E1 trimers form a band around 250 KDa, indicating trimers, but in the blot the complex is clearly much bigger (this is also corroborated by the BN PAGE in Figure 6b, where the size is greater than 400 KDa). This is fortunate as it does not seem likely that E1E2 trimers would run at only 250 KDa. Given the BN PAGE data I would not be surprised if their E2E1 complex is not a "clean" trimer; maybe even a tetramer. Did the authors estimate the expected size of an E2E1 trimer (with the I53-50A)?

L. 176. Results. Missing a "1" in E2E1.

L. 201. Results. Decimal places missing in reference to "r" values.

L. 231. Results. The authors should comment on how they reconcile their finding that E2E1-NP sera competed more with AR3C, when not of the constructs bound AR3C (Figure 1h).

Reviewer #5 (Remarks to the Author):

Dear Authors,

As a Glycoexpert I was only asked to review the Glycan-related part. But in general, the paper is really interesting. It is easy to read and to follow the general experimental setup.

Minor issues:

- missing Supplementary Table 1 legend
- in text (line 329): Figure 5h -> should be 5g

For the glycan-part: the major focus was on the characterization of the E1E2 glycoprotein. It is methodologically sound; no major issues.

Minor issue:

- Figure S3a is blurred and hard to read. For the publication a figure with a higher resolution could be chosen.
- Figure S3b no explanation for the abbreviation of complex glycan composition (A1; FA1;...)

Response to Reviewers for NCOMMS-22-07463 “Induction of cross-neutralizing antibodies by a permuted hepatitis C virus glycoprotein nanoparticle vaccine” by Sliепен et al.

(original reviewers’ comments in *italics*, our response in blue)

Reviewer #1

The authors describe the design of permuted E1E2, its fusion to self-assembling nanoparticles, and the use of this nanoparticle construct as a mosaic vaccine. While the testing of E1E2 antibody binding seems to indicate that the described permuted design does not represent a native-like E1E2 heterodimer, which is disappointing, the results presented regarding the use of the E1E2 design in a nanoparticle and mosaic format should be of interest to the research community. Several concerns, noted below, should be addressed by the authors.

1. It is not clear from the text why the authors refer to their E1E2 design as a “circular permutation” versus a “permutation”. As there are just two proteins (E1 and E2), E2E1 seems to be just a permutation (and the only possible permutation), and the “circular” aspect of the permutation has no meaning in this context. If the authors wish to refer to the design as “circular”, they should explain the meaning of that somewhere in the introduction or results where the design is first mentioned and referred to in that manner. Otherwise, they can just refer to the design as a permutation, to avoid possible confusion for readers.

We regret the confusion that our terminology might have caused. According to Lo et al. NAR 2009: “Circular permutation (CP) in a protein can be considered as if its sequence were circularized followed by a creation of termini at a new location.” Historically, it seems that “circular permutation” refers to domain switching that involves altering the N- and C-termini of a protein (Yu and Lutz Trends in Biotech. 2011) and this includes instances in which the topology of only two domains is switched (e.g. GFP, see Nagai et al. PNAS 2001). Therefore, we referred to our design as being circularly permuted. However, considering this comment and a comment from reviewer #4, we have now changed our manuscript title to “Induction of cross-neutralizing antibodies by a permuted hepatitis C virus glycoprotein nanoparticle vaccine” and changed text accordingly (lines 2; 39; 104; 110; 125-127). We included a reference to “circular permutation” at the start of the results (line 127).

2. The apparent lack of binding to AR4A (Fig. S1e) raises the question of whether the E2E1 assembly designed by the authors resembles native E1E2 or can present any of the known E1E2-dependent antibody epitopes. Even with lack of AR4A binding, other known E1E2 antibodies are available, such as AR5A, or the monoclonals described by Colbert et al. (J Virology 2019), but it is not clear whether the authors tested any antibody outside of AR4A to confirm the nature of the E1E2 assembly. Lack of binding to “certain conformation-dependent bNAbs” is noted on line 377 in the Discussion, suggesting that multiple E1E2 bNAbs were tested instead of a single antibody. Accordingly, the authors should test their design for binding to at least one more E1E2 antibody, such as AR5A, to help readers better understand whether or not the design reflects native E1E2-like antigenicity and can present E1E2-dependent epitopes.

This is a good suggestion. We were unable to obtain AR5A. Instead, as an alternative we have now tested our designs against AT1618, an antibody that is also dependent on native E1E2

folding, similar to AR4A and AR5A (Merat et al. 2019). AT1618 did not bind to our designs. We repeated our ELISAs and have added an additional AT1618 panel in Figure S1f and added lines 155-160 to reflect these additions. These new data reinforce our prior data obtained with AR4A.

3. On lines 136-137, the authors note the “inherent flexibility of the HCV E1E2 glycoprotein” as a possible cause for diffuse E2E1 EM density, however the authors have recently described the high resolution structure of the E1E2 complex (Torrents de la Peña BioRxiv 2021, cited in this manuscript), which indicates that E1E2 is not too flexible for obtaining high resolution structures, contradicting their statement in this manuscript. It seems possible that lack of stability, or added flexibility or dynamics, of the E2E1 design, may be responsible for the diffuse density, and the authors should consider noting this as a possible cause.

E1E2 is known to be highly unstable and it contains highly flexible regions, such as the HVR1 (Stelskja et al 2020) and the CD81 binding site (Kong 2016). As a consequence, solving the structure required a substantial number of tricks to circumvent the instability and heterogeneity. Thus, we first tried to purify E1E2 from the membrane using affinity columns, but this resulted in highly heterogeneous preparations that were unsuitable for structural work. After screening numerous conditions, we found that adding AR4A increased the quality of the protein preparations and that two additional antibodies (AT1209 and IGH505) were needed to obtain preparations of sufficient quality for cryo-EM (Torrents et al. bioRxiv 2021). The E1E2 proteins that were not capable of binding AR4A, AT1209 and IGH505 were excluded from the cryoEM analyses. This suggests that E1E2 is expressed as a highly heterogeneous protein and that complexation with three mAbs including AR4A (or a similar Ab) is needed for generating a homogeneous species amenable for structural analyses. We have now added an additional line to emphasize this point (lines 166-169). In conclusion, we think that these findings are consistent.

4. The lack of binding of the constructs to AR3C is concerning (Figure 1h) and should be addressed by the authors in the Results text. At very least, the E2 control protein would be expected to bind AR3C, and no E2 binding seems to suggest a systematic error with the assay (e.g. a poor-quality antibody), or some properties of the HCV strain’s (AMS0232) E2 and E1E2 sequences that do not permit AR3C binding.

The AMS0232 strain contains several relatively rare mutations in antigenic region 3 (AR3): T431 (~65% D; ~1% T in natural HCV isolates), E434 (~44% N; ~5% E) and I442 (~78% F; ~6% D). Therefore, AMS0232 HCVpp is relatively resistant to several AR3-targeting NAbs, including AR3C (Torrents et al. bioRxiv 2021, Extended Fig. 1B). As stated in the methods (line 717) we introduced I442F, which we did to increase binding of AR3 Abs against our soluble constructs, resulting in AR3B, but not AR3C binding. The lack of AR3C binding was also reason for us to leave out AMS0232 from the mosaic nanoparticle. Note that the other strains, i.e. the six we did select for assembly into mosaic nanoparticles, did show AR3C binding (Figure 4D) ruling out that this is an aberrant result, for example stemming from poor-quality AR3C prep. Rather than a concerning property we think this is just a reflection of HCV genetic and antigenic diversity. We have now more clearly stated the peculiarities of the AMS0232 strain in lines 115-119 and 210-212.

5. The ID50 titers noted in Figure 2 and in the associated Results text seem quite low, and it is not clear whether the quantities noted (e.g. geometric mean titers of 34, 21, and an ID50 cutoff of 40) are reliable for comparisons, given the background that is sometimes observed in HCVpp

assays. The authors should provide information in the Methods regarding the starting serum dilution level used for the neutralization assays (e.g. 1:20 or 1:40), and they should provide example curves from a neutralization assay of immunized and pre-immune rabbit sera (at least one rabbit per group) in a supplemental figure.

Indeed, heterologous titers were low, yielding low geometric mean ID50 titers of 21, 21 and 34 for E2, E2E1-I53-50A and E2E1-NP respectively. We adjusted the text to clarify that 21 is the GMT for E2 and E2E1-I53-50A (lines 255-256). Note that the titers against the sequence-matched AMS0232 HCVpp were much higher for E2E1-NP (median ID50 of 457). Usually, we use VSV-G or MLV controls to test for aspecific neutralization by sera as this provides an internal control for the sera themselves. We now also tested neutralization by week 0 sera against AMS0232 and observed no neutralizing activity. We have added additional Figure S5 that includes the neutralization curves for week 0 and week 6 against AMS0232 HCVpp for all rabbits and the neutralization curves of the same week 6 sera tested against VSV-G controls. These controls show that pre-immunization rabbit sera did not neutralize AMS0232 nor did week 6 sera neutralize the unrelated VSV-G pseudovirus (lines 247-249). This makes us confident that the low titers we measured are reliable. The starting serum dilution for all our neutralization assays was 1:20 and this has now been mentioned in the methods section (lines 944-945).

6. It is not clear why the authors selected the permuted version of E1E2 (E2E1) over the non-permuted version for their nanoparticle and in vivo studies, at least based on the antigenicity data shown in Figure S1. Although the hypothetical proximity of the E2 C-terminus to the E1 N-terminus is noted by the authors as a rationale for the permuted design, it still seems theoretically possible that E1E2-I53-50A would be antigenic and would form nanoparticles. The authors should comment on why the selection of E2E1 was made for nanoparticle fusion, or show the data that were used to support that.

We regret that the rationale was not entirely clear. At the start of this project, we based our assumptions on the locations of the termini of E1 and E2 on the Giang et al. PNAS 2012 and Gopal et al. PLoS Path 2017 papers, which suggested that the E2 C-terminus and E1 N-terminus are in close proximity, providing impetus for our permuted E2E1 design. Another assumption was that E1E2 trimerizes through the E1 transmembrane domains (Falson 2015). Therefore, switching E1 and E2 and subsequently fusing E1 to the I53-50A trimerization domain appeared to be a logical strategy to us. We have now added a sentence to clarify this point. In parallel, we also tried to solve the structure of native membrane-bound E1E2 (Torrents et al. bioRxiv 2021). This structure confirmed that the E1 C-terminus is more distant from N-terminus of E2 (i.e. E1E2) than the E2 C-terminus from the N-terminus of E1 (i.e. E2E1). We have now added lines 130-140 in the Results text and Discussion to solidify the rationale for our choice. We also added Figure S1c to illustrate the distance between the E2 C-terminus and E1 N-terminus.

7. At least two previous studies have reported permuted E1E2 constructs (Prentoe et al. 2021, Guest et al. 2021, cited in this study). The authors do not seem to discuss these E2E1 precedents; they should be noted by the authors (along with E2E1 precedent studies, if applicable), and briefly compared with the currently described E2E1 design.

The reviewer rightly notes that two other studies appeared while our work was ongoing. We did mention these papers in the Discussion earlier (lines 396-398, and added the recent Wang et al. PNAS 2022 reference), but we have now extended this by adding a section comparing the differences between these designs (lines 411-418). We note that the Prentoe and Guest studies

did not present the permuted E1E2 constructs on nanoparticles and the E2E1 construct in the Prentoe study did not improve immunogenicity compared to E2

Reviewer #2

The aim of this study was to examine the potential of nanoparticles comprised of the E1/E2 glycoproteins of HCV to elicit bNAb in vaccinated rabbits. In the past, this has proved to be a difficult as the virus shows high sequence variability in these envelope proteins which are the target of NAb.

The study was well conceived, the manuscript was well written and supported by an appropriate bibliography and comprehensive figures.

The study is the result of a considerable amount of work, not least the glycan shield studies and builds on a previous study of the E1/E2 structure of the virus by the authors to apply the 153-50NP technology used previously in COVID-19 and RSV vaccine studies to HCV.

We thank the reviewer for the positive feedback.

Several points should be addressed;

Lines 39 et seq; As some current licensed vaccines use a cocktail and as a number of research groups are investigating the use of a cocktail for different infectious agents, I believe that the abstract should reflect the interesting data viz. that the mosaic was more effective than a cocktail in eliciting bNAb.

We have now rephrased the abstract somewhat with more focus on the mosaic nanoparticle result. We indeed observed a trend suggesting that the mosaic nanoparticle groups elicited a broader response than the cocktail nanoparticle group. However, the difference was not significant, so we would refrain from claiming that the mosaic group is superior to the cocktail group in the abstract.

Lines 66 et seq; It is not clear if the authors equate an established infection with a persistent infection. In my view, and according to the general dogma, cell mediated immunity is generally required to clear an established (persistent) infection, and earlier studies from the Chisari group demonstrated this very well. As there is no evidence for ADCC as a potential mechanism for clearance of HCV-infected hepatocytes, the authors should dilute the impact of NAb on clearance and introduce a (important) role for cellular immunity in this process.

We have now toned down this section and indicated the importance of T cell immunity, and added the Chisari references (Neumann-Haefelin et al. JCI 2005, Thimme et al. JEM 2001) (lines 69-71).

Lines 176 et seq; As I read the text and compared the text in this section with the figures, I noted that the authors failed to mention any comparison between the E2E1-NP and the E2E1-15350A trimers until line 208, essentially as a throw-away line. I kept asking why the authors failed to present this comparison, and as I'm certain that other readers will think similarly, I believe that this is an important comparison that should be given greater emphasis in the text from line 222.

We regret that our initial manuscript was unclear with respect to this comparison. We have now added additional text to detail the difference between E2E1-I53-50A and E2E1-NP and E2 with regards to E2E1, E2 and HVR1 binding titers (in lines 224-225, 227-229, 238-240, 242), AMS0232 neutralization (245-247) and AMS3a E2E1-foldon binding (line 278) and the normalized data shown in Figure S6 (lines 294-297) (old Figure S4).

Lines 280 et seq; what was the basis for choosing UKN4.1.2 NPs for the B cell activation studies?

We chose UKNP4.1.1 because it showed relatively strong binding to AR3C in BLI compared to mosaic E2E1-NPs (Figure 4d and new Figure S9). We added a sentence to explain our rationale for choosing UKNP4.1.1 (lines 339-340).

Minor points;

Line 51; suggest “persist” for “linger”

This has been modified accordingly.

Line 106; insert “the TMD and” immediately prior to “the predicted membrane-proximal”

Likewise, this has been modified.

Line 118; insert “the particles” immediately prior to “purified”

We have now inserted “the proteins” prior to “purified” (currently line 144).

Figure legends; in figures which show the NS-EM images, the white bar is not mentioned in the legends until figure 4 which does not contain NS-EM images

We have now indicated the size (200 nm) in each of the images that contain a scale bar.

Reviewer #3

Although direct-acting antiviral (DDAs) can cure most hepatitis C virus (HCV) infections, they cannot prevent liver damage and reinfection. Development of a broadly protective HCV vaccine thus remains a top priority. In this manuscript, Sliepen et al. presented an HCV vaccine strategy that displays soluble E1E2 dimers derived from one strain, or different genotypes (mosaic), on a two-component nanoparticle (I53-50) as vaccine candidates. The authors then immunized rabbits to compare the neutralizing antibody (NAb) responses induced by E2, E2E1-I53-50A trimer, and E2E1-NP. The results showed moderate improvement of breadth and potency. Overall, this is an interesting study carried out by a group of vaccine experts. However, despite the novelty of the vaccine design, there are a lot of concerning issues that may not be easily addressed. Some self-conflicting data or statements need to be clarified or corrected during revision.

We thank the reviewer for acknowledging the interest and novelty of our study. We have addressed his/her concerns and clarified various statements according to the reviewer's requests.

Major comment #1:

There is not enough evidence to support the notion that HCV glycoprotein is a homotrimer of E1E2 heterodimer like HIV-1 Env, and HCV is not a class-I fusion virus. The Falson study only had low-resolution (SDS-PAGE) data, whereas the Freedman study was based on computational modeling only. The authors' statements on page 4, lines 79-83, seem to be self-contradictory. If they already solved a high-resolution structure of native E1E2 dimer on the cell membrane, how can this data be used to support E1E2 being a trimer on the virion surface? Are there any studies on the E1E2 multimeric state when displayed on cell versus virus surface?

We regret the confusion that the earlier version of our manuscript might have caused. The current study was started shortly after the Falson 2015 and Freedman 2017 studies were published. We took their results into account when designing our constructs and therefore used I53-50A to trimerize E1E2. It was not until later that we managed to solve the structure of membrane-bound E1E2 (Torrents et al. bioRxiv 2021) showing that E1E2 forms a heterodimer. However, we cannot rule out that E1E2 forms trimers on the viral surface, which would be in line with the Falson and Freedman studies, although the trimer form might also represent a different conformational state (e.g. an intermediate or postfusion state). Irrespective of whether E1E2 naturally forms trimers, trimerizing a soluble E1E2 heterodimer has three potential advantages. First, it hides the membrane-facing non-neutralizing side of E1E2. Second, it increases valency, which increases B cell receptor cross-linking and B cell activation. Third, it facilitates incorporation into icosahedral nanoparticles, further boosting immunogenicity. Furthermore, we found that E2E1-I53-50A presents a more complete glycan shield than E2 and this might help to direct NAb responses to the right epitopes. We have now clarified these points in the Results (lines 124-142) and Discussion (419-445).

Major comment #2:

On page 4, lines 107-108, and page 5, 109-112, the authors designed soluble E1E2 and E2E1 and selected E2E1 for further development based on the Alanine-scanning work of Giang et al. and Gopal et al. It is somewhat puzzling why the authors used such low-resolution data, and not their high-resolution cryo-EM structure, for this most critical decision in construct design. As the top experts in structure-based vaccine design, the authors need to provide some good explanations. The authors are also encouraged to confirm, based on their cryo-EM data, that the N terminus of E1 is indeed close to the C terminus of E2, which is the basis of their E2E1 design. Finally, the authors inserted the same 15-aa GS linker in both E1E2 and E2E1 constructs, contradicting their statement that they picked the E2E1 design based on the Alanine-scanning data.

We designed our constructs, performed characterization and immunized the rabbits before we solved the structure of membrane-bound E1E2, which we managed to do only very recently (see response to comment #1). For the design, we utilized the available literature (e.g. Falson et al. J Virol. 2015; Giang et al. PNAS 2012; Gopal PloS Pathog 2017; Freedman et al. J. Virol 2017) and removed the most hydrophobic C-terminal parts of E1 and E2. When the structure was solved, we were pleased to find that the C terminus of E2 is indeed interacting with the N-terminus of E1 providing further encouragement for permuting E1 and E2 for soluble HCV glycoprotein design. The distance of the E2 C-terminus at D698 to the E1 N-terminus Y196 is ~18 Å. The linker length was chosen because we did not have the structure yet and figured we

needed a relatively long linker to allow proper folding. Comment #5 from reviewer #1 was similar and in response we have added lines 128-135 in the Results and added Supplemental Figure S1c to show the distances between the termini.

Major comment #3:

On page 5, lines 133-138, the authors first claimed a homogeneous E1E1-I53-50A trimer based on the NS-EM data and then said that they were unable to get a higher-resolution reconstruction due to flexibility. First, the same data cannot be used to support opposite conclusions depending on what the authors need. Second, since the authors have determined a high-resolution cryo-EM structure for the full-length, native E1E2 dimer, they should have more than enough expertise to get a reasonable NS-EM model. Third, if flexibility is indeed the problem, does that mean further optimization of the linker length is required at this point?

We agree with the reviewer that our statements might seem contradictory and we changed lines 161-166 and added lines 166-169. We also added lines 459-461 in the part of the Discussion that involves improvements to our design, and now includes optimization in linker length. For the apparently paradoxical observations on instability and heterogeneity versus the ability to obtain a (homogeneous) structure, see our response to reviewer #1 point 3.

Major comment #4:

On page 5, lines 167-168 (Figure 1h), the authors measured the antigenicity of three constructs against a panel of NAbS. However, AR4A, a well-known bNAb that only recognizes the native E1E2 dimer, was not included in the antigenic evaluation. It was then stated on page 13, lines 376-378, that “We realize that our current E2E1-I53-50A design probably does not represent the native conformation of viral E1E2, because it does not bind certain conformation-dependent bNAbS, such as AR4A (Giang et al., 2012)”. This is a fundamental problem of this study.

Over the last decade, the same group of authors emphasized the importance of native-like Env trimers in the HIV vaccine field and coined the terms “pseudotrimers” or TINOs (trimer-in-name-only) to describe those Env antigens that cannot bind conformation-dependent bNAbS such as PGT145. The lack of AR4A binding suggest that E1E1-I53-50A suffers a similar problem to those HIV pseudotrimers and TINOs. According to the authors’ many reviews in the HIV field, such non-native antigens should not be used in vaccine development.

Another issue is that the authors already reported a high-resolution cryo-EM structure for the native E1E2 dimer in complex with AR4A. It is puzzling why in this study they would move forward with a problematic antigen design when they could easily access the correct answer. The authors need to provide convincing explanations for all these choices.

We fully realize that the lack of AR4A-binding indicates that the HCV E1E2 protein is not entirely ‘native-like’ and have stated so in the text as the reviewer points out. We also acknowledge that native-like immunogens are probably necessary as components in a vaccine regimen that induces HIV-1 bNAbS. However, HCV is not HIV and we do not yet know what is required for inducing HCV bNAbS by vaccination. The level of ‘nativeness’ required in a vaccine might very well differ substantially per virus. For example, in the case of RSV, only stabilized pre-fusion trimers are capable of inducing potent neutralizing antibodies, while non-native (post-fusion) trimers are not. On the contrary, unstabilized SARS-CoV-2 Spike trimers (as in the AstraZeneca vaccine which lacked the proline and other mutations to stabilize Spike that are present in the Pfizer, Moderna, Novavax and Janssen vaccines) and Spike fragments (i.e. RBD, the receptor binding domain) are readily able to induce neutralizing antibodies

against SARS-CoV-2. Our HCV E1E2 protein presents the CD81 receptor binding site (i.e. AR3) very efficiently and we hypothesized that it would be able to induce antibodies against this site (i.e. AR3-directed antibodies: see Figure 3f). Our immunization studies clearly indicate that the construct is superior at inducing neutralizing antibodies compared to state-of-the-art constructs. Might there be further room for improvement with more native-like immunogens? We believe so, in particular when considering inducing AR4A-like antibodies. We are actively working in this area using the new structural information. This has also been mentioned in the discussion section (lines 459-461).

As to the question why we did not use the high-resolution structure for our design, the answer is quite straightforward. We simply did not have the structure that could have acted as a design template at that time. These permuted constructs were designed in 2018 and we finally succeeded in solving the structure in 2021. Instead, we used the available literature to generate an E1E2-based construct that can be expressed (in contrast to mostly E2 designs that were available at the time). We had already started generating mosaic NPs and immunization studies in rabbits by the time we finally resolved the long-sought after E1E2 structure. Importantly the structure confirmed that the C- and N-termini of E2 and E1 are in close proximity providing post-hoc justification for the permuted design to generate novel soluble native-like E1E2 proteins (see response to comment #2 and Figure S1c), although we acknowledge that improvements might be possible by exploiting the now available structural information.

It is extremely exciting to now have the structure of membrane-bound E1E2 (Torrents et al. bioRxiv 2021), as it will catalyze structure based immunogen improvement. We have now clearly stated that the E1E2 structure was solved after we initiated the current study lines (lines 128-133) and provided some discussion regarding potential design improvements (lines 459-461).

Major comment #5:

On page 7, lines 180-181, the authors created an E2E1-foldon probe for analyzing E2E1-directed binding antibody responses in vaccination. The same structural and antigenic evaluation must be done for E2E1-foldon to ensure that the same E2E1 conformation is presented on I53-50A and foldon, otherwise, the response measured by this probe would be biased.

The reviewer raises a fair point. We have now tested the antigenicity of E2E1-foldon in ELISA and show the comparison with E2E1-I53-50A in Supplemental Figure S4b. We conclude that the antigenicity is very similar between the two.

Furthermore, the use of an E2E1 probe to analyze rabbit serum will no doubt result in a much lower reading for the E2 group and favor the two vaccine antigens that contain both E1 and E2. The authors should include an E2 probe in the analysis, which could provide insight into: (1) the impact of multivalent display on antibody titer (since all three antigens contain E2), and (2) the effect of E1 on E2-directed antibody response, in both a trimeric form and an NP form.

We did measure anti-E2 responses using a specific E2 probe (Figure 2C), and we found that binding responses in the E2 immunized rabbits were significantly lower than in animals receiving E2E1 trimers or E2E1-NP (lines 226-229).

Major comment #6:

On page 7, 193-197, the authors reported enhanced HVR1-specific antibody responses for E2E1-NP, raising the concern that the multivalently displayed HVR1 will distract immune system from more conserved epitopes such as the CD81-binding loop, front layer, and AS412 that are targeted by bNAbs. Although the mosaic E2E1-NP design was intended to reduce the HVR1-specific antibody response by diversifying its sequence, the concern is that a large repertoire of weak or non-neutralizing HVR1 antibodies would be generated by such mosaic vaccines.

This is a very good point and one that we considered extensively before we selected the E1E2 proteins to be co-assembled into the mosaic nanoparticles. As pointed out in the results (lines 309-316), we actually used the mosaic nanoparticle setup to “dilute” the HVR1 response by deliberately selecting E1E2 proteins with maximally diverse HVR1 domains, and were quite successful in doing so.

A monotypic E2E1-NP, presents 60 copies of HVR1 (20 E2E1 trimers with each 3 HVR peptides). In contrast, a mosaic E2E1-NP containing 6 different E2E1 trimers presents only on average 10 copies of each HVR1 peptide, while 60 copies of cross-reactive bNAb epitopes are presented. Indeed, binding titers measured against a mix of the 6 HVR1 peptides showed a trend ($p=0.15$, added now in line 358) that HVR1 binding is lower than those raised by the cocktail vaccine and comparable to what was raised by the monovalent vaccine (Fig. 5b). And were significantly lower when taking into account the binding ratio of HVR1 vs. the E2E1 trimers (Figure 5b), indicating lower HVR1 responses when corrected for E2E1 trimer binding. Future design considerations might involve removing the HVR1 altogether, although this has yielded limited success so far (Law et al. J. Virol 2020, Prentoe et al. PloS ONE 2021). We discuss these design choices in the Discussion section (lines 470-478).

Major comment #7:

On page 9, lines 246-249, the authors reported an interesting finding that the AS412 and AR3C epitopes contributed to the increased neutralizing breadth seen for the E2E1-NP group. This begs the question whether an E1E2 or E2E1 dimer-based vaccine strategy is truly superior to an E2-based vaccine strategy, because the AS412 and AR3C epitopes are on the E2 surface and remain intact in all E2 core designs after truncation of HVR1, VR2, and VR3. Based on this finding, the authors may want to discuss the different goals of the dimer and E2-based vaccine strategies.

The reviewer raises an important point and we have now added a section on the different goals of HCV glycoprotein vaccine strategies (lines lines 470-478, see also our response to comment #6 in the Discussion), which incorporates some of the points raised in comment #6. AP33 and AR3C partly overlap and emerging evidence suggests that a vaccine eliciting antibodies against multiple non-overlapping antigenic regions is probably preferred over a vaccine eliciting NAb against a single antigenic region (Kinchen et al. JCI 2019). Therefore, an antigen that presents multiple non-overlapping neutralizing epitopes (e.g. (improved versions of) E2E1) might be preferred over antigens that present only one (E2 core).

Major comment #8:

On page 10, lines 273-275, the authors missed E2E1-I53-50A trimers of 6 genotypes with I53-50B to form mosaic NPs. Since a fully assembled I53-50 NP can present 20 trimers and there are 6 different E2E1-I53-50A trimers, the final mosaic E2E1-NP vaccine may be a mixture of many forms with different genotype ratios, structural layouts, and melting temperatures. How can such a mosaic vaccine be characterized and pass a rigorous QC system in GMP

manufacturing? The authors need to discuss these potential caveats related to mosaic NP vaccines.

Indeed, the stochastic assembly of mosaic I53-50 nanoparticles results in a normal distribution, leading to the heterogeneity of nanoparticles, which might be challenging from the translational perspective. Our aim was to provide proof-of-concept at the preclinical level. Our results and those obtained by others in the field of influenza (Kanekiyo et al. Nat. Immunol. 2019; Boyoglu-Barnum et al. Nature 2021) and COVID (Cohen et al Science 2021, Cohen et al Science 2022), should provide the impetus to develop manufacturing protocols that pass rigorous QC testing for GMP manufacturing. Furthermore, nanoparticle systems with hierarchical assembly that result in homogeneous mosaic nanoparticles are also under development. We added a section in the discussion to highlight these challenges and opportunities (lines 484-496).

Major comment #9:

Discussion, page 12-14, needs to be revised to include objective and balanced discussions about the issues listed in #1-8. For example, on page 14, lines 407-409, it is claimed that “the geometry of E2E1 on I53-50 NPs suitably presents the desired cross-neutralizing epitopes”. However, on the previous page 13, lines 376-378, there is a statement that “it doesn’t bind certain conformation-dependent bNAbs, such as AR4A”. These two statements contradict each other. The native-like E1E2 construct reported by Guest et al. is clearly more suitable for NP vaccine development.

We have adjusted the discussion section (detailed in our responses to comments #1-8) and changed the seemingly contradictory statement to “suitably presents these E2-specific cross-neutralizing epitopes” (line 504, with line 500 added for context).

Major comment #10:

On page 14, lines 409-412, the authors mentioned the COVID-19 vaccine based on the I53-50 NP platform as supporting evidence for their HCV E2E1-I53-50 NP vaccine. Some web search revealed that this COVID-19 vaccine (Walls et al., 2020) was licensed to a company, Icosavax, which conducted a phase-I/II trial with early data released on March 25. The results were “below the company’s expectations” and “discordant with the preclinical data” published in Cell (Walls et al., 2020). In fact, the vaccine response appeared to be “comparable to or below the placebo”. In some reports, it was suggested that the platform has a “fatal flaw”. In light of this new human data on the I53-50 NP platform, the authors still need to discuss the potential implications and how this would affect the translational path for their HCV vaccine based on the same platform.

We thank the reviewer for raising this important point, while we also emphasize the risks of citing the (rather sensationalist) biotech press instead of peer-reviewed scientific articles. The reviewer is correct that IVX-411, an I53-50-based nanoparticle vaccine displaying the receptor binding domain (RBD) of the SARS-CoV-2 spike, developed by Icosavax, reported unexpectedly poor results in a first Phase I/II trial. However, this isolated result does not indicate a "fatal flaw" in the platform, as the biotech press suggested. In fact, in November 2021, SK bioscience, a vaccine manufacturer in South Korea developed the same nanoparticle vaccine with support from the Bill & Melinda Gates Foundation (BMGF) and the Collaborative for Epidemic Preparedness Initiative (CEPI), and reported potent immunogenicity from their I53-50-based vaccine, GBP510 in humans (https://www.skbioscience.co.kr/en/news/news_01_01?mode=view&id=99&page=2). Those data are online in a preprint on medRxiv

(<https://www.medrxiv.org/content/10.1101/2022.03.30.22273143v1>) and the peer-reviewed version was recently published in eClinicalMedicine ([https://www.thelancet.com/journals/eclinm/article/PIIS2589-5370\(22\)00299-1/fulltext](https://www.thelancet.com/journals/eclinm/article/PIIS2589-5370(22)00299-1/fulltext)), and supported advancing GBP510 into a multinational Phase 3 trial. In late April 2022, SK announced that GBP510 met its co-primary endpoints in Phase 3, showing superior immunogenicity and seroconversion rates to the AstraZeneca comparator vaccine (https://www.skbioscience.co.kr/en/news/news_01_01?mode=view&id=125&). SK has recently received regulatory approval for use of its GBP510 vaccine in South Korea and will follow up with filings in other jurisdictions https://www.skbioscience.co.kr/en/news/news_01_01?mode=view&id=132&. These data suggest that Icosavax's initial results are not representative for the I53-50 platform, and clearly establish that I53-50-based vaccines can be potent and safe in humans. We have added lines 504-508 to the Discussion to clarify this to the reader.

Reviewer #4

The authors describe a novel two-component vaccine approach for making an HCV vaccine. The approach is interesting as it allows for easy incorporation of higher order forms of the HCV envelope proteins (here E2E1 trimers most likely), which cannot be easily accommodated in “standard” one-component VLP vaccine approaches such as previously described for HCV. Although the overall improvement using the mosaic E2E1-NPs is modest it appears that the conclusion that this two-component vaccine merits further study is sound. In addition, the authors have performed an impressive array of antigen/immunogen characterizations, which also has relevance for HCV vaccine development. This includes glycan analyses of purified protein variants and their extensive testing of mosaic or cocktail NPs, which unfortunately yielded very modest improvements compared with a single isolate E2E1-NP. Their data that E2E1 trimers represents a promising HCV vaccine antigen is less convincing. The initial comparison with what they refer to as “monomeric E2” does not show that E2E1 trimers are in fact superior. In fact, it seems the opposite may well be true (see below).

We thank the reviewer for asserting that E2E1-NPs merit further study. We would not refer to the improvements as “modest”, especially when compared to ‘state-of-the-art’ monomeric E2 (see our specific responses below).

They attempt to compare serum antibodies derived from mice immunized with lower order E2 complexes (probably not monomers, and certainly a mix of E2 species; see Figure S6b) with E2E1 (which form >400 KDa complexes, maybe trimers) and even larger E2E1-NPs. Not surprisingly, E2 does not induce the same levels of E2 or E1E2 antibodies, which is likely due to its smaller size. The authors themselves find that overall antibody levels correlate with neutralization (Figure 2f). Moreover, the authors report that anti-E2E1 antibodies are elevated 60 fold for the E2E1-NPs compared with E2, whereas neutralization is only increased 10-fold (l. 240; Results). This indicates that while the antibody levels are, unsurprisingly, increased on E2E1 and E2E1-NPs (partly due to size and increased avidity of epitopes), the quality of those antibodies is decreased (around 6 fold). Their epitope exposure/binding tests using Octet also shows that while E2E1 binds less to several non-neutralizing MAbs it also has reduced binding to AR3B, AT12009, AT12011 and HC84.26). In fact, the only E2 antibody that binds more strongly to E2E1 trimers is the AP33, which targets a linear epitope. Taken together this may indicate that E2E1 does not fold properly, which is also in line with the lack of AR4A binding, and likely folds “worse” than E2 alone (at least from a vaccine

antigen perspective). A proper control would be to purify a single species of E2 (monomer, dimer or something else) and then attach this to their NP and test immunogenicity in comparison with E2E1-NPs.

For these reasons I disagree with several of the author's key claims as presented both in the Results section as well as in the Discussion.

Below we have addressed the specific points that are summarized by the reviewer above.

In addition, other major/minor points of interest/concern are listed below:

Major general issues:

Generally, I think the authors could spend a bit more time briefly describing how the methods they use allow them to address the questions they ask. As it is now a reader would have to be quite knowledgeable in a wide range of scientific methodology to easily read the manuscript (see also some of the minor specific issues for cases where this is particularly problematic).

We regret that the use of a wide range of scientific methodologies has led to unclarity to scientists less familiar with these techniques. We have now explained more carefully why a particular methodology was used throughout the paper (e.g. lines, 175-177, 195-200)

Major specific issues:

It is not possible to see the E2 band in Figure S1d. Unless the size of E2 is obscured by the E1 band (which the authors do not mention) this is unexpected. It is also of some concern as it is possible that E2 is partially lost to the complex once furin cleavage occurs. A repeat with more protein, clearly showing both E1 and E2 preferably with WB data as well, for both proteins would be a very relevant and simple control to perform.

We thank the reviewer for addressing this. In our previous experiment, we loaded 3 µg of each protein, resulting in lower absolute amounts of E2. We have now repeated the experiments while loading equimolar E2 in each lane (e.g. 4 µg E2, 6 µg E2E1 and 8.2 µg E2E1-I53-50A) (new Figure S1d) showing that E2 is not lost, but that the apparent loss observed previously can be attributed to the amount of protein loaded in the sample lane.

L. 131. Results. The authors do not see binding of AR4A to the potential E2E1 trimers. As the authors acknowledge in the Discussion, this likely indicates that E2E1 trimers are not in a native E1E2 conformation with resulting differences in binding for some MAbs. It would be relevant to test binding of other MAbs known to require an E1E2 interaction for binding, such as AR5A or HEPC146.

Reviewer #1 had a similar comment (comment #1). See our reply there. In summary, we have tested our proteins using AT1618, an antibody with similar properties as AR4A, AR5A and HEP146 (Merat 2019 J. Hepatology). We did not have AR5A or HEPC146 at our disposal. The data obtained with ART1618 reinforce our prior results with AR4A.

Minor general issues:

It seems and oversight on behalf of the authors not to mention the two Christiansen et al. papers on an HCV E2 VLP vaccine (2018; 2019, Sci Rep).

We thank the reviewer for pointing out these relevant papers, which we have now included in our Introduction (line 103) and Discussion (lines 409-410).

The use of the term “circular” permutation is not explained. I am guessing that it simply refers to the swapping of E1 and E2 in the primary coding sequence, but the authors may mean something different. It should be explained clearly.

Considering a similar comment from Reviewer #1 (comment #1) we have now removed the “circular” from our title and text. We mention that such a permutation is sometimes referred to as circular permutation (line 126-127).

The abbreviation HILIC—UPLC is used without spelling out the acronym in the main text. Also, I think it would be helpful to general readers with a line or two concerning what the technique accomplishes (i.e. how it enables glycan analysis).

We spelled out the abbreviation and added two sentences on how this method enables glycan analysis (lines 173-177)

The authors refer to BLI, but make no mention of it in the method section, leaving it up to the readers to know (or guess) that the Octet is in fact a BLI instrument. The BLI data is also shown only as relative intensities in a color table, making it difficult to assess how substantial the differences really are. A supplementary figure with the actual binding curves would be very nice.

The curves from the experiments have now been added (Supplemental Fig. 4a and 5) and we now clarified the text by removing the word Octet throughout the text and only referring to biolayer interferometry (BLI). Furthermore, we changed the heatmap color scheme of Fig. 1h and Fig. 4d to allow better distinction between binding differences.

The authors NPs that are populated with several different E2E1 trimers (mosaic NPs). However, it does not seem like they have any measure of whether the different E2E1 trimers are bound to the NPs at the same frequency (i.e. the E2E1 stoichiometry on the particles). Is there some way to measure this?

This is a good point that we have considered as well. Determining the stoichiometry could in theory be done using novel mass-spectrometry methods coupled with negative or positive selection using specific antibodies, but this is technically very challenging, in particular when six different proteins are used, as we did here. However, we have performed such mass spectrometry measurements using an I53-50-based mosaic nanoparticle displaying two coronavirus Spike proteins: those of SARS-CoV and SARS-CoV-2 (Brinkkemper et al. submitted). This particular experiment with only two different proteins involved several months of trial-and-error and required considerable time to optimize. Importantly, the results indicated stochastic assembly of the mosaic NPs. Using more than two antigens on the same nanoparticles would require a disproportionate amount of effort, while there is no reason to assume that mosaic nanoparticle assembly would not be stochastic. The immunogenicity results indicate that the mosaic nanoparticles actually worked as intended and induced a substantially different

antibody response compared to a cocktail of monovalent NPs. In light of the reviewer's comment, we have added some discussion on this topic (lines 484-496).

The authors claim that the mosaic E2E1 NPs induce less HVR1-specific antibodies, but that is not clear from the 6-valent HVR1 ELISAs they perform. It is good they normalize the HVR1-specific antibody responses against the overall antibody response. However, they do not find differences for mosaic E2E1-NPs compared with AMS3a-NPs, which is not surprising as antibodies against other HVR1 sequences (present in the mix) will obscure any reduction in AMS3a HVR1-specific antibodies. They should supplement this by running an AMS3a HVR1-specific ELISA for the same mouse sera as shown in Figure 5b.

We thank the reviewer for this suggestion and have now added the AMS3a HVR1 and E2E1-foldon-His ELISA binding data the reviewer requests. We observed a trend that monovalent AMS3a E2E1-NPs induced higher titers of antibodies recognizing AMS3a E2E1-foldon and HVR1 compared to mosaic E2E1-NPs, although this was not statistically significant. These data are now added as Figure S10 and lines 349-352.

Minor specific issues:

I could not find a specific reference to the sequence and truncation sites of soluble E2. The assumption would be that it is truncated at 698 and of the AMS3032 isolate, but this should be made clear both in the Methods and in the Results section, where the E1E2 and E2E1 constructs are explained.

This has now been added in the Results section and we rephrased the Method section (lines 143; 721-723).

L 52. Intro. I do not think it is clear, or even likely, that long term ill-effects of HCV infection is linked to hepatocyte killing. Long-term liver inflammation is believed to play a major role in this. As it is not central to the study itself, I recommend taking out the information regarding the way in which HCV leads to HCC and other liver-related disease.

We agree this was too speculative and we have now removed this statement from the introduction.

L 93. Intro. "Sofar" as two words.

This has been fixed

L. 94. Intro. Soluble E1E2 trimers should be E2E1 trimers.

This has now been fixed

L. 126. Results. The authors state that the E2E1 trimers form a band around 250 KDa, indicating trimers, but in the blot the complex is clearly much bigger (this is also corroborated by the BN PAGE in Figure 6b, where the size is greater 400 KDa). This is fortunate as it does not seem likely that E1E2 trimers would run at only 250 KDa. Given the BN PAGE data I would not be surprised if their E2E1 complex is not a "clean" trimer; maybe even a tetramer. Did the authors estimate the expected size of an E2E1 trimer (with the I53-50A)?

The proteins indeed migrate somewhat slower than might be expected. Based on protein alone, E2E1-I53-50A trimer should be 231 kDa, but this calculation disregards glycans. Assuming ~15 glycans per E2E1 protomer of on average ~2 kDa this would add around 90 kDa in size. Therefore, in theory, this would mean that a trimer would run at ~320 kDa. However, it is well-established that glycans alter (usually decrease) the electrophoretic mobility of glycoproteins making PAGE analyses less reliable for estimating the sizes of glycoproteins. Furthermore, E2E1-I53-50A efficiently assemble into full nanoparticles and this would only be possible if the antigens are trimeric. We changed “around 250 kDa” in “larger than 250 kDa” (line 153).

L. 176. Results. Missing a “1” in E2E1.

This has now been corrected.

L. 201. Results. Decimal places missing in reference to “r” values.

This has been corrected.

L. 231. Results. The authors should comment on how they reconcile their finding that E2E1-NP sera competed more with AR3C, when not of the constructs bound AR3C (Figure 1h).

These immunizations were performed using AMS3a E2E1-I53-50NPs, which does bind to AR3C, in contrast to AMS0232-based E2E1-NPs (see Figure 4d).

Reviewer #5

Dear Authors,

As a Glycoexpert I was only asked to review the Glycan-related part. But in general, the paper is really interesting. It is easy to read and to follow the general experimental setup.

We thank the reviewer for the positive feedback.

Minor issues:

- missing Supplementary Table 1 legend

The legend has now been added.

- in text (line 329): Figure 5h -> should be 5g

This has now been fixed.

For the glycan-part: the major focus was on the characterization of the E1E2 glycoprotein. It is methodologically sound; no major issues.

We are happy to hear that the reviewer asserts that our methods are sound.

Minor issue:

-Figure S3a is blurred and hard to read. For the publication a figure with a higher resolution could be chosen.

We have now supplied a higher resolution version.

- Figure S3b no explanation for the abbreviation of complex glycan composition (A1; FA1;...)

This has now been added in the legend of Figure S3 with an added reference to a more detailed overview in Behrens et al. Cell Rep. 2016.

REVIEWERS' COMMENTS

Reviewer #1 (Remarks to the Author):

The authors have sufficiently addressed all of my comments.

Reviewer #3 (Remarks to the Author):

Review of manuscript NCOMMS-22-07463A

The authors have addressed most of this and other reviewers' comments and suggestions. The manuscript has been substantially revised to reflect these changes. However, the reviewer is still conflicted about the manuscript. There is a large body of work behind this manuscript, and HCV vaccine is certainly important. Multivalent display of HCV E1E2 (or E2E1 in this case) dimers on a two-component nanoparticle is novel, at least from a technical point of view. However, as detailed in the reviewer's previous comments, the antigen design was fundamentally flawed and inconsistent with their own cryo-EM model, the SARS-CoV-2 vaccine based on this nanoparticle platform scored poorly in humans (completely failed in the US trial – a formal investigation was launched and concluded antigen instability – and showed mediocre responses in the South Korea trial – 2-3 fold better than the AZ vaccine means still worse than mRNA vaccines, inconsistent with results from other protein vaccines such as Novavax and Clover vaccines), and their animal studies showed little advantage of the designed vaccine constructs.

At the same time, this manuscript represents the first such study in the field and provides critical insights into whether this type of constructs will advance HCV vaccine development. Although the answer is likely a big NO, the information, and particularly the lessons, may still be valuable to the HCV vaccine community and the broad readership of this journal.

Reviewer #4 (Remarks to the Author):

With the exception of the comments below, all comments have been satisfactorily addressed.

Excerpt from rebuttal letter(I have added my response at the end of each paragraph):

The authors describe a novel two-component vaccine approach for making an HCV vaccine. The approach is interesting as it allows for easy incorporation of higher order forms of the HCV envelope proteins (here E2E1 trimers most likely), which cannot be easily accommodated in "standard" one-component VLP vaccine approaches such as previously described for HCV. Although the overall improvement using the mosaic E2E1-NPs is modest it appears that the conclusion that this two-component vaccine merits further study is sound. In addition, the authors have performed an impressive array of antigen/immunogen characterizations, which also has relevance for HCV vaccine development. This includes glycan analyses of purified protein variants and their extensive testing of mosaic or cocktail NPs, which unfortunately yielded very modest improvements compared with a single isolate E2E1-NP. Their data that E2E1 trimers represents a promising HCV vaccine antigen is less convincing. The initial comparison with what they refer to as "monomeric E2" does not show that E2E1 trimers are in fact superior. In fact, it seems the opposite may well be true (see below)

Author response:

We thank the reviewer for asserting that E2E1-NPs merit further study. We would not refer to the improvements as "modest", especially when compared to 'state-of-the-art' monomeric E2

(see our specific responses below).

Reviewer response:

It is not a reasonable comparison to compare a small monomeric protein (sE2) with a large protein, especially when presented on a nanoparticle. This has also been shown for HCV (Vietheer 2017 Hepatology). Additionally, several groups have shown that VLPs/NPs enhance antibody responses to HCV envelope proteins. See specific point below.

They attempt to compare serum antibodies derived from mice immunized with lower order E2 complexes (probably not monomers, and certainly a mix of E2 species; see Figure S6b) with E2E1 (which form >400 KDa complexes, maybe trimers) and even larger E2E1-NPs. Not surprisingly, E2 does not induce the same levels of E2 or E1E2 antibodies, which is likely due to its smaller size. The authors themselves find that overall antibody levels correlate with neutralization (Figure 2f). Moreover, the authors report that anti-E2E1 antibodies are elevated 60 fold for the E2E1-NPs compared with E2, whereas neutralization is only increased 10-fold (l. 240; Results). This indicates that while the antibody levels are, unsurprisingly, increased on E2E1 and E2E1-NPs (partly due to size and increased avidity of epitopes), the quality of those antibodies is decreased (around 6 fold). Their epitope exposure/binding tests using Octet also shows that while E2E1 binds less to several non-neutralizing MAbs (e.g. AR3B, AT12009, AT12011 and HC84.26) it also has reduced binding to AR3B, AT12009, AT12011 and HC84.26). In fact, the only E2 antibody that binds more strongly to E2E1 trimers is the AP33, which targets a linear epitope. Taken together this may indicate that E2E1 does not fold properly, which is also in line with the lack of AR4A binding, and likely folds "worse" than E2 alone (at least from a vaccine antigen perspective). A proper control would be to purify a single species of E2 (monomer, dimer or something else) and then attach this to their NP and test immunogenicity in comparison with E2E1-NPs.

For these reasons I disagree with several of the author's key claims as presented both in the Results section as well as in the Discussion.

Author response:

Below we have addressed the specific points that are summarized by the reviewer above.

Reviewer response:

Having carefully combed through the author responses mentioned above I have not been able to find answers to the concerns I raised above. In the case that we are "speaking" past one another I will re-iterate below:

The authors are correct that the E2E1-foldon construct on VLPs is superior to sE2 in inducing antibodies and it is also true that sE2 and its variants are state-of-the-art (although that is likely changing). However, I am not able to find any data that supports that E2E1-foldon is a superior antigen to sE2. In fact, as stated below in the initial review it seems very likely that the effect is a result of an overall increase in induction of antibodies, due to size and perhaps increased avidity. Thus, it seems unclear whether attaching sE2 on the NPs would have worked as well or better than attaching sE1E2-foldon proteins. This possibility is supported by the ELISA data showing decreased reactivity of E2E1-foldon to (conformational) epitopes compared with sE2 (see above for additional details). In the absence of a compelling reason as to why the E2E1-foldon protein is superior to sE2, the authors should tone down this aspect in the discussion or, at least, mention the ELISA and neutralization data they present which could indicate that it isn't the case.

Reviewer #1 (Remarks to the Author):

The authors have sufficiently addressed all of my comments.

Reviewer #3 (Remarks to the Author):

Review of manuscript NCOMMS-22-07463A

The authors have addressed most of this and other reviewers' comments and suggestions. The manuscript has been substantially revised to reflect these changes. However, the reviewer is still conflicted about the manuscript. There is a large body of work behind this manuscript, and HCV vaccine is certainly important. Multivalent display of HCV E1E2 (or E2E1 in this case) dimers on a two-component nanoparticle is novel, at least from a technical point of view. However, as detailed in the reviewer's previous comments, the antigen design was fundamentally flawed and inconsistent with their own cryo-EM model, the SARS-CoV-2 vaccine based on this nanoparticle platform scored poorly in humans (completely failed in the US trial – a formal investigation was launched and concluded antigen instability – and showed mediocre responses in the South Korea trial – 2-3 fold better than the AZ vaccine means still worse than mRNA vaccines, inconsistent with results from other protein vaccines such as Novavax and Clover vaccines), and their animal studies showed little advantage of the designed vaccine constructs.

At the same time, this manuscript represents the first such study in the field and provides critical insights into whether this type of constructs will advance HCV vaccine development. Although the answer is likely a big NO, the information, and particularly the lessons, may still be valuable to the HCV vaccine community and the broad readership of this journal.

Reviewer #4 (Remarks to the Author):

With the exception of the comments below, all comments have been satisfactorily addressed.

Excerpt from rebuttal letter(I have added my response at the end of each paragraph):

The authors describe a novel two-component vaccine approach for making an HCV vaccine. The approach is interesting as it allows for easy incorporation of higher order forms of the HCV envelope proteins (here E2E1 trimers most likely), which cannot be easily accommodated in “standard” one-component VLP vaccine approaches such as previously described for HCV.

Although the overall improvement using the mosaic E2E1-NPs is modest it appears that the conclusion that this two-component vaccine merits further study is sound. In addition, the authors have performed an impressive array of antigen/immunogen characterizations, which also has relevance for HCV vaccine development. This includes glycan analyses of purified protein variants and their extensive testing of mosaic or cocktail NPs, which unfortunately yielded very modest improvements compared with a single isolate E2E1-NP. Their data that E2E1 trimers represents a promising HCV vaccine antigen is less convincing. The initial comparison with what they refer to as “monomeric E2” does not show that E2E1 trimers are

in fact superior. In fact, it seems the opposite may well be true (see below)

Author response:

We thank the reviewer for asserting that E2E1-NPs merit further study. We would not refer to the improvements as “modest”, especially when compared to ‘state-of-the-art’ monomeric E2 (see our specific responses below).

Reviewer response:

It is not a reasonable comparison to compare a small monomeric protein (sE2) with a large protein, especially when presented on a nanoparticle. This has also been shown for HCV (Viethier 2017 Hepatology). Additionally, several groups have shown that VLPs/NPs enhance antibody responses to HCV envelope proteins. See specific point below.

They attempt to compare serum antibodies derived from mice immunized with lower order E2 complexes (probably not monomers, and certainly a mix of E2 species; see Figure S6b) with E2E1 (which form >400 KDa complexes, maybe trimers) and even larger E2E1-NPs. Not surprisingly, E2 does not induce the same levels of E2 or E1E2 antibodies, which is likely due to its smaller size. The authors themselves find that overall antibody levels correlate with neutralization (Figure 2f). Moreover, the authors report that anti-E2E1 antibodies are elevated 60 fold for the E2E1-NPs compared with E2, whereas neutralization is only increased 10-fold (l. 240; Results). This indicates that while the antibody levels are, unsurprisingly, increased on E2E1 and E2E1-NPs (partly due to size and increased avidity of epitopes), the quality of those antibodies is decreased (around 6 fold). Their epitope exposure/binding tests using Octet also shows that while E2E1 binds less to several non-neutralizing MAbs it also has reduced binding to AR3B, AT12009, AT12011 and HC84.26). In fact, the only E2 antibody that binds more strongly to E2E1 trimers is the AP33, which targets a linear epitope. Taken together this may indicate that E2E1 does not fold properly, which is also in line with the lack of AR4A binding, and likely folds “worse” than E2 alone (at least from a vaccine antigen perspective). A proper control would be to purify a single species of E2 (monomer, dimer or something else) and then attach this to their NP and test immunogenicity in comparison with E2E1-NPs. For these reasons I disagree with several of the author’s key claims as presented both in the Results section as well as in the Discussion.

Author response:

Below we have addressed the specific points that are summarized by the reviewer above.

Reviewer response:

Having carefully combed through the author responses mentioned above I have not been able to find answers to the concerns I raised above. In the case that we are “speaking” past one another I will re-iterate below:

The authors are correct that the E2E1-foldon construct on VLPs is superior to sE2 in inducing antibodies and it is also true that sE2 and its variants are state-of-the-art (although that is likely changing). However, I am not able to find any data that supports that E2E1-foldon is a superior antigen to sE2. In fact, as stated below in the initial review it seems very likely that the effect is a result of an overall increase in induction of antibodies, due to size and perhaps increased avidity. Thus, it seems unclear whether attaching sE2 on the NPs would have worked as well or better than attaching sE1E2-foldon proteins. This possibility is supported by the ELISA data showing decreased reactivity of E2E1-foldon to (conformational) epitopes compared with sE2 (see above for additional details). In the absence of a compelling reason as to why the E2E1-foldon protein is superior to sE2, the authors should tone down this aspect in the discussion or, at least, mention the ELISA and neutralization data they present which could indicate that it isn't the case.

The immunogenicity results in Fig 2 and Fig 3 (and summarized as normalized data in Fig S6) consistently show a trend that indicates that E2E1-I53-50A improves (N)Ab responses compared to E2. Additionally, Fig. 2h, and Fig 3i suggest that rabbits immunized with E2E1-I53-50A trimers elicited broader responses and that the elicited serum Abs seemed to compete with a broader range of bNAb epitopes (Fig. 3f). Together, this indicates that also the quality of the humoral response was improved despite the differences in antigenicity (Fig. 1h). However, we cannot rule out that the increased immunogenicity is (partly) due to the increased quantity of anti-E2 antibodies elicited by E2E1-I53-50A. Therefore, we have now added a section in the discussion on the possibility that the quality of the humoral response elicited by E2E1-I53-50A is not necessarily improved (lines 406-409).